# Genetic defects in β-spectrin and tau sensitize *C. elegans* axons to movement-induced damage via torque-tension coupling

Michael Krieg[1,2†], Jan Stühmer[3], Juan G Cueva[1], Richard Fetter[1], Kerri Spilker[4‡], Daniel Cremers[3], Kang Shen[4], Alexander R Dunn[2], Miriam B Goodman[1*]

[1]Department of Molecular and Cellular Physiology, Stanford University, Stanford, United States; [2]Department of Chemical Engineering, Stanford University, Stanford, United States; [3]Department of Informatics, Technical University of Munich, Germany; [4]Department of Biology, Stanford University, Stanford, United States

**Abstract** Our bodies are in constant motion and so are the neurons that invade each tissue. Motion-induced neuron deformation and damage are associated with several neurodegenerative conditions. Here, we investigated the question of how the neuronal cytoskeleton protects axons and dendrites from mechanical stress, exploiting mutations in UNC-70 $\beta$-spectrin, PTL-1 tau/MAP2-like and MEC-7 $\beta$-tubulin proteins in *Caenorhabditis elegans*. We found that mechanical stress induces supercoils and plectonemes in the sensory axons of spectrin and tau double mutants. Biophysical measurements, super-resolution, and electron microscopy, as well as numerical simulations of neurons as discrete, elastic rods provide evidence that a balance of torque, tension, and elasticity stabilizes neurons against mechanical deformation. We conclude that the spectrin and microtubule cytoskeletons work in combination to protect axons and dendrites from mechanical stress and propose that defects in $\beta$-spectrin and tau may sensitize neurons to damage.

*For correspondence:
mbgoodmn@stanford.edu

Present address: [†]ICFO, Institute of Photonic Sciences, Barcelona, Spain; [‡]Biogen, Cambridge, United States

## Introduction

Neurons play a central role in receiving and distributing information and depend on long, slender axons and dendrites for their function. Vertebrate peripheral neurons can be meters long and less than 1 μm in diameter: the peripheral neurons of Cetaceans have aspect ratios exceeding a factor of 10,000,000 (*Wedel, 2012*). Despite their slender construction, peripheral neurons have axons that bend without breaking—that is they are robust enough to withstand the mechanical deformations generated during body movements under most conditions. Conversely, injuries that lead to compression of the longest human nerve, the sciatic nerve, can cause severe back pain and palsy. The neurons of the central nervous system are also subject to mechanical deformations during contact sports or other activities, despite the protection afforded by the skull (e.g. *Broglio et al., 2011*). In all neurons, mechanical trauma can disrupt the cytoskeleton and trigger axonal degeneration (reviewed by *Gaetz, 2004*). Thus, it is of central importance to understand how neurons maintain their shape and function in the mechanically active environment of our bodies.

Axons harbor bundles of microtubules (MT), which are known to be important for neuronal architecture, growth, and organelle transport (*Tyler, 2012*). MTs are needed for resistance to compressive loads in fibroblasts and cardiac myocytes, especially when the MTs are laterally constrained (*Brangwynne et al., 2006*). Considering this, it is likely that MTs also affect how axons respond to mechanical loads. For instance, *Tang-Schomer et al. (2010)* showed that rapid mechanical stretch

deforms cortical axons grown on elastic substrates and that the observed deformations were enhanced by treatment with the MT stabilizing drug, taxol, and suppressed by treatment with the MT stabilizing drug, nocodazole. Taxol slowed and nocodazole accelerated stretch-induced axonal degeneration, suggesting that intact MTs are needed to protect neurons from mechanical strain-induced damage

Microtubules are not the only components of the neuronal cytoskeleton. Actin and spectrin assemble into a subcortical lattice with a ~200 nm periodicity in neurons and glial cells (*D'Este et al., 2016*, *2015*; *He et al., 2016*; *Xu et al., 2013*; *Zhong et al., 2014*). Spectrin filaments are composed of α and β subunits, which associate head to head to form tetramers: the N-terminal end of each α-subunit associates with the C-terminus of each β-subunit. In axons and dendrites, spectrin filaments have an estimated length of 200 nm, a value larger than expected from the their resting contour length (*Boal, 1999*). Genetic defects leading to a reduction of tension in the actin-spectrin networks cause *C. elegans* neurons to buckle and deform during body movements, leading to axon damage (*Hammarlund et al., 2007*; *Krieg et al., 2014*).

Here, we use *C. elegans* to study how neurons maintain stable shapes even under the large mechanical strains generated by body movements. We focused on the six touch receptor neurons (TRNs) required for sensing gentle touch (*Chalfie et al., 1985*) because they are among the best characterized neurons in the *C. elegans* nervous system. TRNs have a simple morphology and long neurites that emanate from the cell body and tile the body into anterior/posterior and ventral/lateral dermatomes. The anatomy of *C. elegans* TRNs has been studied intensively through light- and electron microscopy and their neurites contain >60 MT cross-linked into a coherent bundle (*Chalfie and Thomson, 1979*; *Cueva et al., 2007*). The TRN microtubules depend on expression of *mec-7* β-tubulin and *mec-12* α-tubulin and together these genes account for more than 90% of the tubulin transcripts expressed in these neurons (*Lockhead et al., 2016*). TRNs also contain an actin-spectrin network, in which spectrin is under constitutive mechanical tension and has been implicated in mechano-protection (*Krieg et al., 2014*).

Here, we examine how the mechanical interplay between these two cytoskeletal elements influences neuronal shape dynamics and resistance to mechanical stress by leveraging mutations in UNC-70 β-spectrin, MEC-7 β-tubulin, and in PTL-1 tau, the sole *C. elegans* homolog of MAP2/tau proteins. Our observations indicate that mechanical neuroprotection depends on spectrin-dependent longitudinal tension, MT-dependent bending stiffness, and MT-binding proteins such as PTL-1 that act to dissipate torsional forces that would otherwise destabilize MTs and the neuron (*Grason, 2015*; *Grason and Bruinsma, 2007*). Because spectrin, β-tubulin, and tau are highly conserved from *C. elegans* to humans and constitute the major protein content in the mammalian central nervous system, our results provide a mechanistic framework for understanding how neurons remain resilient in the face of mechanical stress.

## Results

Mechanical stability is one of several vital functions provided by the neuronal cytoskeleton, which includes both actin-spectrin networks linked to the plasma membrane and bundles of microtubules (MTs) that fill the cytoplasm. To understand how these structures interact to protect axons and dendrites from mechanical stress in living animals, we examined axon shape in mutants carrying genetic defects in the constituents of actin-spectrin networks and MT bundles. Our study builds on prior work in this area: Genetic defects in UNC-70 β-spectrin have been linked to movement-dependent axon fracture (*Hammarlund et al., 2007*) and buckling (*Krieg et al., 2014*) in *C. elegans* motor neurons and TRNs, respectively. Additionally, morphological distortions in the TRNs such as bends and kinks have been described in animals carrying loss-of-function alleles of *mec-17* α-tubulin acetyltransferase (*Topalidou et al., 2012*) and missense alleles of *mec-12* α-tubulin (*Hsu et al., 2014*). No such shape distortions appear in mutants lacking MT bundles in the TRNs (*Chalfie and Thomson, 1982*; *Hsu et al., 2014*; this study—*Figure 1A*), suggesting that defects in MT bundles, but not their absence, cause shape distortions

To learn more about the interaction between the spectrin and the microtubule cytoskeleton, we analyzed TRN shape in mutants affecting the microtubule cytoskeleton and asked whether or not these mutations could enhance shape defects in *unc-70(e524)* missense mutants. The *e524* allele encodes a missense mutation, E2008K, affecting the linker region between the tetramerization

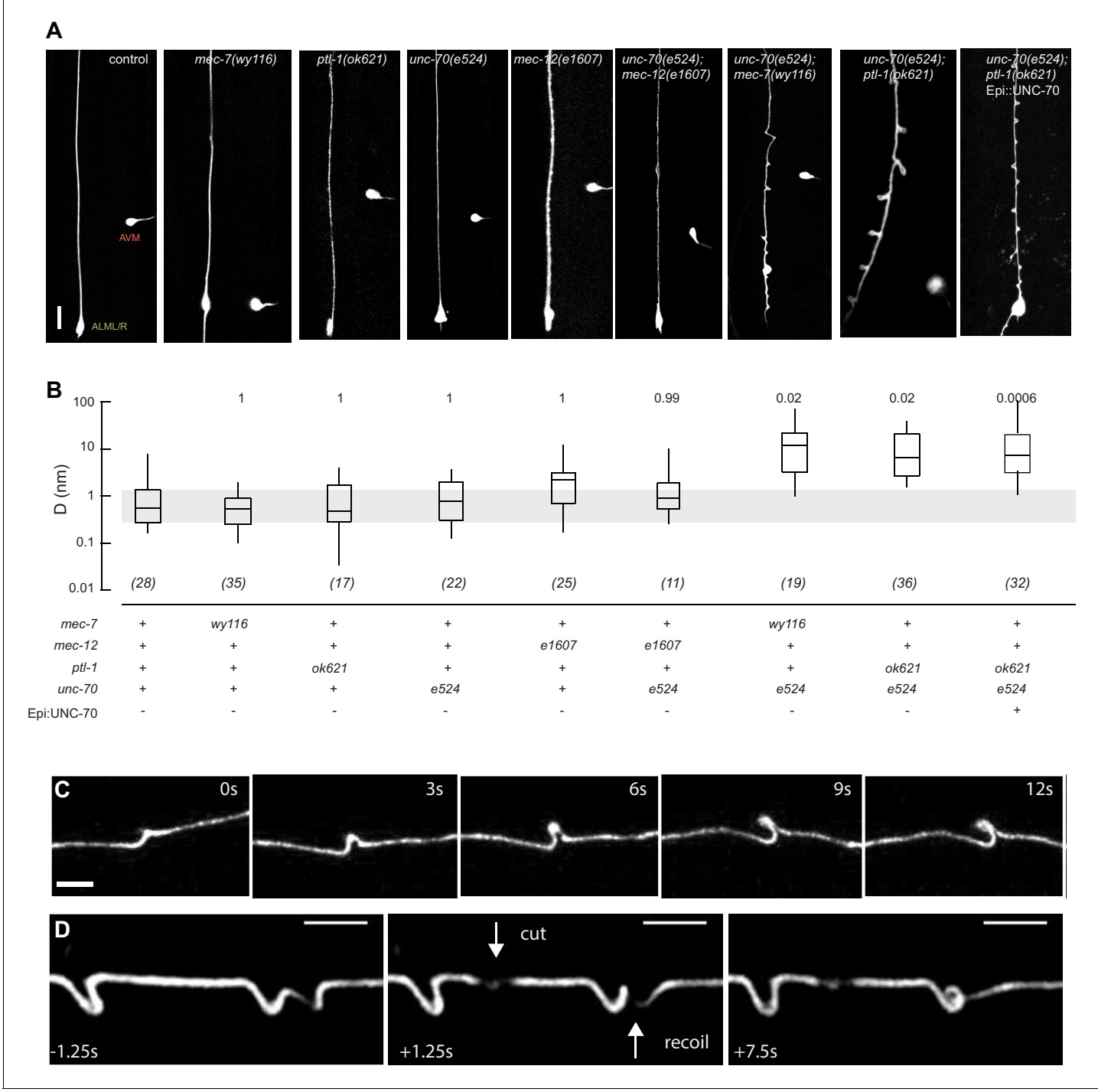

**Figure 1.** Mutations in the spectrin and microtubule cytoskeletons cause shape defects in *C. elegans* TRNs. (**A**) Representative images of ALM in control and the indicated genotypes; anterior is up and ventral is to the right. All neurons are marked by TRN::GFP encoded by the *uIs31[mec-17p::GFP]* transgene. Scale = 10 μm. (**B**) Quantification of ALM shape as local neuron randomness (Methods) in the indicated genotypes. Numbers in parentheses indicate number of animals analyzed and the numbers above each box are p-values derived from a multi-comparison Tukey posthoc *t*-test. Exact p-values were calculated according to Newman-Keuls. (**C**) Still images from *Video 2* showing the dynamics of the defects in *unc-70(e524);ptl-1(ok621)* double mutants. (**D**) Still images of *Video 6* showing the dynamics of the defects after stress release by laser cutting. *Videos 1–6* are associated with the analyses and still images represented in *Figure 1*.

The following figure supplements are available for figure 1:

*Figure 1 continued on next page*

*Figure 1 continued*

**Figure supplement 1.** Alignment of the C-terminal tail of three major *β*-tubulin isoforms expressed in *C. elegans* TRNs (MEC-7, TBB-1, and TBB-2) and human neurons (TUBB-2,–3, −4) (*Leandro-García et al., 2010*).

**Figure supplement 2.** Molecular model of an α*β*-tubulin heterodimer (*Ravanbakhsh et al., 2013*) with the *mec-7(wy116[T409I])* substitution shown in space filling representation (PDB accession code 1Z2B).

domain and the 16th spectrin repeat of *β*-spectrin (*Krieg et al., 2014*). To enable quantitative comparisons, we devised a method to quantify the neurite's deviation from the straightest possible path. In brief, we measured the deviation perpendicular to the long (anterior-posterior) axis of the animal and the axon at all positions and reduced these measures to a single value, *D*, which describes the displacement of the neurite about this axis on a pixel-by-pixel basis. In this computation, *D* has a value of zero for a perfectly straight neuron and any deviations from straightness increase *D*, regardless of the nature of the morphological defect (see Materials and methods). Using this approach, we examined a novel allele of *mec-7 β*-tubulin, *wy116*, and a null allele of *ptl-1* tau, *ok621*.

The *wy116* allele replaces threonine 409 with an isoleucine (*Figure 1—figure supplement 1*) and maps to a region in the C-terminal tail of MEC-7 *β*-tubulin thought to form the interface between α and *β* tubulins (*Figure 1—figure supplement 2*). The *wy116[T409I]* allele was isolated in a genetic screen for defects in synaptic vesicle transport (Materials and methods) and has a mild defect in touch sensation (*Table 1*). As in wild type, TRNs in young-adult *mec-7(wy116)* animals were straight and lacked local bends and shape distortions (*Figure 1A and B*). The TRNs in *mec-7(wy116);unc-70(e524)* double mutants displayed large bends and kinks that were not found in either single mutant. Such shape defects were largely absent in *mec-12(e1607); unc-70(e524)* double mutants (*Figure 1A and B*) that lack MEC-12 α-tubulin and fail to form 15pf MTs in TRNs (*Chalfie and Thomson, 1982*). This finding and the fact that *wy116* affects a region implicated in binding MT-associated proteins (MAPs) including tau (*Al-Bassam et al., 2002*; *Littauer et al., 1986*) led us to examine mutants lacking PTL-1, the sole tau homolog in the *C. elegans* genome (*Goedert et al., 1996*; *Gordon et al., 2008*).

As found in *mec-7(wy116)*, TRNs in *ptl-1(ok621)* null mutants were straight, lacked obvious shape distortions, and had *D* values close to zero (*Figure 1A and B*). In *ptl-1(ok621);unc-70 (e524)* double mutants, by contrast, the TRN axons exhibited large bends and localized helices,

**Table 1.** Mean touch response. Animals assayed as young adults, grown at 20°C (except for strains carrying *mec-7(e1343)* which were grown at 25°C) and blind to genotype in cohorts of ~25 animals. Except for *ptl-1(ok621);unc-70(e524);uIs31*, at least two independent cohorts were tested for each genotype. There was a significant effect of genotype (one-way ANOVA, $F_{(9, 1019)}=305.3$, $p<0.0001$) and the indicated genotypes differed from control at $p<0.05$ (*), $p<0.001$ (***) level according to a Dunnett's posthoc comparison (ns = denotes no significant difference).

| Genotype | Touch response (*mean*±*SD*) | # of worms |
|---|---|---|
| control (*uIs31[mec-17p::GFP]*) | 9.35 ± 1.15 | 75 |
| *unc-70(e524);uIs31* | 6.33 ± 2.12* | 203 |
| *ptl-1(ok621);uIs31* | 8 ± 1.4* | 50 |
| *ptl-1(ok621);unc-70(e524);uIs31* | 5.3 ± 2.65*** | 75 |
| *ptl-1(ok621);unc-70(e524);mec-7(e1343);uIs31* | 1.5 ± 1.2*** | 25 |
| *mec-7(wy116);unc-70(e524);uIs31* | 4.3 ± 0.77*** | 50 |
| *mec-7(wy116);uIs31* | 8.6 ± 0.6[ns] | 142 |
| *mec-7(wy116);ptl-1(ok621);uIs31* | 9.5 ± 0.3[ns] | 154 |
| *mec-7(e1343);uIs31* | 1.2 ± 1.3*** | 137 |
| *ptl-1(pg73[ptl-1::mNeonGreen])* | 9.6 ± 0.77[ns] | 50 |

loops, and plectonemes (*Figure 1A*, *Video 1*). Given that rescue of β-spectrin in epidermal cells reduced but did not eliminate buckling instabilities in AVM neurons (*Krieg et al., 2014*), we examined *ptl-1(ok621);unc-70(e524)* double mutants expressing wild-type UNC-70 β-spectrin in the epidermis from the *oxIs95[epi::UNC-70]* transgene. This manipulation resulted in a less pronounced phenotype but did not obviously alter the displacement metric, *D* (*Figure 1A*). The defects in *ptl-1;unc-70* mutants appeared at roughly regular intervals of 13 ± 0.5 μm (mean ± s. e.m., *n* = 121 intervals in 10 neurons) along the axon (*Figure 1*) and were dynamic. *Figure 1C* is a series of images illustrating the dynamic nature of the shape defects seen in *ptl-1;unc-70* double mutant axons (see also *Video 1* and *Video 2*). The shape defects in *ptl-1;unc-70* double mutants were qualitatively similar, but more pronounced than those observed in *mec-7(wy116); unc-70* double mutants (*Figure 1A*).

The defects we observed in living animals are reminiscent of what happens following the partial release of tension in an elastic rope that is twisted and held under tension. The dynamics of this situation are familiar to anyone handling twisted ropes or threads (*Video 3*) and are well-described by a physical model in which the formation of coils and plectonemes is the result of an imbalance between torque, tension, and bending stiffness (*Coyne, 1990*). This physical model also predicts that no coils form in the absence of twist (*Video 4*). Based on these observations, we hypothesized that the shape defects observed in mutant axons reflected the same physical process. If the analogy holds, then releasing axial tension in mutant axons via in vivo laser axotomy is expected to cause rotation along the axon's long axis and conversion of twist into writhe evident in the formation of new coils and plectonemes. Consistent with this prediction, releasing axial tension via in vivo laser axotomy readily drove reorganization of coils and plectonemes (*Figure 1D*, *Video 5*). In some cases, we also found evidence of rotation following axotomy (*Video 6*), indicative of a release from a pre-existing, twisted state.

## The position of axons within tissues determines their exposure to mechanical stress

Mechanical stresses generated by tissue movement are not expected to be uniform throughout the tissue. The worm's cylindrical body and the differential position of the lateral (ALML, ALMR, PLML, PLMR) and ventral (AVM, PVM) TRNs provide an opportunity to investigate this question. Given that *C. elegans* crawl on their sides and bend primarily along their dorsal-ventral axis, the lateral midline constitutes the animal's neutral bending axis (*Figure 2A*). Euler beam bending theory (*Bauchau and Craig, 2009*) predicts that strain, ε, within the worm's cylindrical body is the product of the

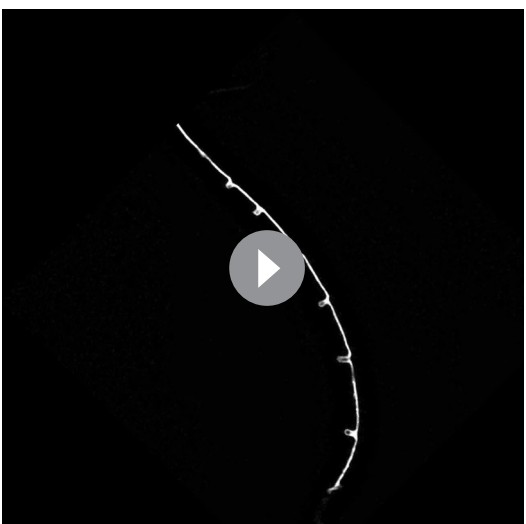

**Video 1.** Representative movie related to *Figure 1* showing a GFP-labeled ALM neuron in an *unc-70(e524); ptl-1(ok621)* mutant background.

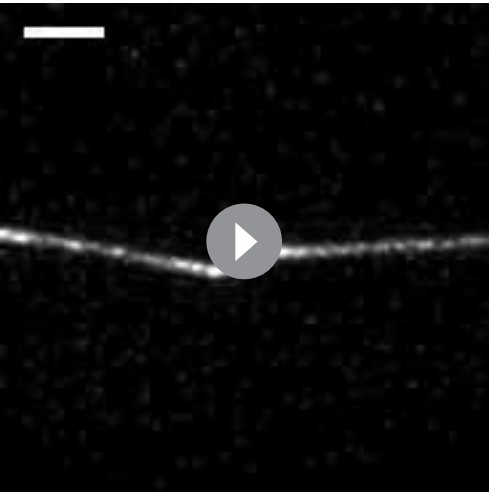

**Video 2.** Dynamics of a single axonal defect in a moving *unc-70(e524); ptl-1(ok621)* animal. Related to *Figure 1*.

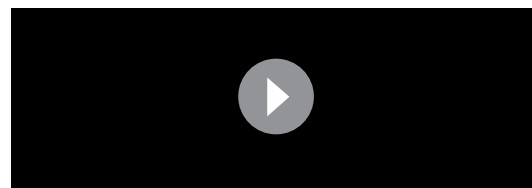

**Video 3.** A twisted elastic nylon thread is held under tension. Upon reduction of the end-end distance (by an increase in slack or reduction in tension), the initially straight thread first forms a continuous spiral, which then deforms into a localized, superhelical defect, called a plectoneme. Related to *Figure 1*.

curvature, $c$, of the neutral bending axis and the displacement, $z$. away from the neutral bending axis: $\varepsilon = cz$. In wild-type control animals (*Figure 2A*), the AVM axon is located along the ventral midline at a distance of $z = 21.1 \pm 6.5$ μm (mean ± s.d., $n = 32$ worms) and the ALM axons lie much closer to the neutral bending axis at $z = 5.5 \pm 1.7$ μm ($n = 15$ worms). If Euler's simple theory of mechanical stress in a bending cylindrical beam holds for *C. elegans*, then the body bends that occur during movement are expected to deform the AVM axon four-fold more than the ALM axon.

Consistent with this idea and prior work (*Krieg et al., 2014*), wild-type AVM axons undergo local contractions and extensions with up to 50% strain observed during normal locomotion (*Figure 2B and C*, *Figure 2—figure supplement 1*; *Video 7*). The ALM axons, by contrast, show only modest changes in strain during movement (*Figure 2B and C*; *Figure 2—figure supplement 1*; *Video 8*). As an additional test of the idea that the larger displacement of AVM from the neutral bending axis is responsible for the larger change in strain, we took advantage of a mutation in *him-4* hemicentin that displaces the ALM away from the neutral bending axis and toward the dorsal side of the animal (*Figure 2D*) but has little effect on AVM's position (*Vogel and Hedgecock, 2001*). Thus, in *him-4* mutants AVM lies along the ventral midline and ALM lies near the dorsal midline at $z = 17 \pm 2.6$ μm (mean ± s.d., $n = 9$ worms, *Figure 2D and E*). Measuring the relationship between neuron strain and body strain in *him-4* mutants, we found that both AVM and ALM axons generate similar local contractions and extensions during locomotion (*Figure 2E and F*; *Figure 2—figure supplement 1*; *Video 9*, *Video 10*). These findings suggest that AVM and ALM axons have similar mechanical properties and that axon position has dramatic effects on bending-induced mechanical stresses.

Next, we sought to elucidate the interplay of axon position and the tension born by the spectrin cytoskeleton. Previously (*Krieg et al., 2014*), we showed that *unc-70 β*-spectrin mutant neurons have reduced axial tension and that the axons of AVM neurons in *unc-70* mutants undergo buckling instability under movement-induced compressive, but not tensile stress (*Figure 3A,B and C*; *Video 11*). However, similar defects are not observed in the axons of ALM neurons in these mutants (*Figure 3A,B and C*, *Video 12*). The simplest explanation for this discrepancy is that because the ALM axons are positioned close to the animal's neutral bending axis, they are exposed to significantly smaller compressive stresses during body bending. We exploited the fact that the ALM axons are displaced away from the neutral bending axis in *him-4* mutants (*Figure 3D*) to test this model. Indeed, both AVM and ALM exhibited buckling during locomotion-induced compression in *him-4; unc-70* double mutants, consistent with the large strains experienced by both neurons in this animal

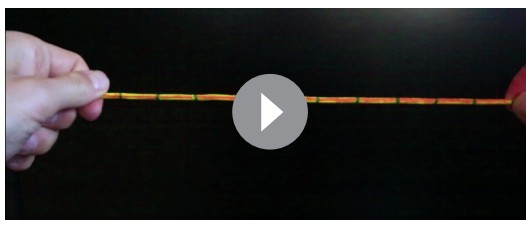

**Video 4.** An untwisted elastic nylon thread is held under tension. The thread remains straight until the tension is released. Upon further reduction of the end-end distance (effective compression), the thread begins to buckle with a single arc, reminiscent of classical Euler buckling. Related to *Figure 1*.

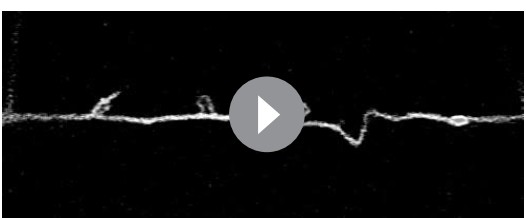

**Video 5.** Example of a laser axotomy that caused rotation along the axon's longitudinal axis. Cut occurs at the 10th movie frame. Related to *Figure 1*.

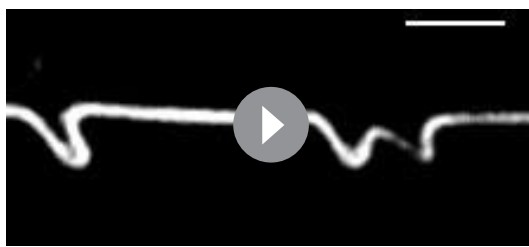

**Video 6.** Example of a laser axotomy that triggered the transformation of a localized helix into a loop after cutting. Cut occurs at the third movie frame. Related to *Figure 1*.

(*Figure 3E and F*; *Figure 3—figure supplement 1*; *Video 13*, *Video 14*). Collectively, these findings support the idea that peripheral axons are subjected to mechanical stresses in a manner that depends on their anatomical position within the tissue.

## A balance of tension and torque governs axonal resilience to mechanical stress

We sought to develop a quantitative model describing how mechanical stability of neuronal processes might arise in vivo. To do so, we applied Euler-Kirchhoff theory describing the time and space evolution of a filament subjected to bending, stretching and twisting stresses (*Bergou et al., 2010*; *Goyal et al., 2005*; *Nizette and Goriely, 1999*; *Purohit, 2008*; *Van der Heijden and Thompson, 2000*). The key factors in the model, which we implemented by modeling axons as discrete elastic rods (DER model), are mechanical parameters that measure the energetic cost of bending, stretching, and twisting the axon. They are bending stiffness $B$ (*Equation 1*), axial stiffness $k$ (*Equation 2*), and twist rigidity $C$ (*Equation 3*):

$$B = EI \tag{1}$$

$$k = EA/L \tag{2}$$

$$C = 2GI \tag{3}$$

where $E$ is the Young's modulus, $I$ is the second moment of area of the cross section, $A$ is the cross-sectional area, $L$ is the length, and $G$ is the shear modulus. The twist rigidity $C$ depends on the shear modulus $G$ (*Equation 4*), because twisting generates shear stresses when two neighboring cross-sections are rotated in respect to another. The relation between $G$ and $E$ is set by the Poisson's ratio $\nu$:

$$G = \frac{E}{2(1+\nu)} \tag{4}$$

To obtain initial values for the relevant mechanical parameters, we estimated the Young's modulus from elasticity measurements of cultured, GFP-tagged TRNs by indenting them with an atomic force microscope (*Figure 4—figure supplement 1*). In parallel, we used genetic and pharmacological perturbations to evaluate the predicted contributions of the actin-spectrin and microtubule cytoskeleton to Young's modulus, $E$. Using this approach, we found that wild-type axons were stiffer ($E = 6.3 \pm 0.7$ kPa, mean $\pm$ sem, $n = 5$) than those treated with the actin depolymerizing agent Latrunculin A ($E = 4.9 \pm 1$ kPa, $n = 5$; p=5·10$^{-4}$, Mann-Whitney U-test, *Figure 4—figure supplement 2*). Similar results have been reported for mammalian sensory neurons (*Magdesian et al., 2012*; *Ouyang et al., 2013*), underscoring the conservation of material properties among neurons. Estimated values for $E$ in *unc-70(e524)* $\beta$-spectrin mutant axons ($3 \pm 1.1$ kPa, mean $\pm$ sem, $n = 4$, p=2·10$^{-3}$) were lower than untreated wild-type axons and indistinguishable from Latrunculin A-treated wild-type axons (p=0.28). We repeated these measurements and compared wild-type axons ($E = 6.1 \pm 0.3$ kPa, mean $\pm$ sem, $n = 11$) to those of *ptl-1(ok621)* ($E = 4.6 \pm 0.6$ kPa, mean $\pm$ sem, $n = 18$) and *mec-7(wy116)* neurons ($E = 4.3 \pm 0.4$ kPa, mean $\pm$ sem, $n = 17$) and found that both classes of mutant axons had significantly lower Young's moduli, in agreement with our hypothesis that mutations affecting the microtubule cytoskeleton lead to decreases in bending rigidity. Collectively, these results demonstrate that axonal stiffness depends on both the actin-spectrin cytoskeleton and MT bundles.

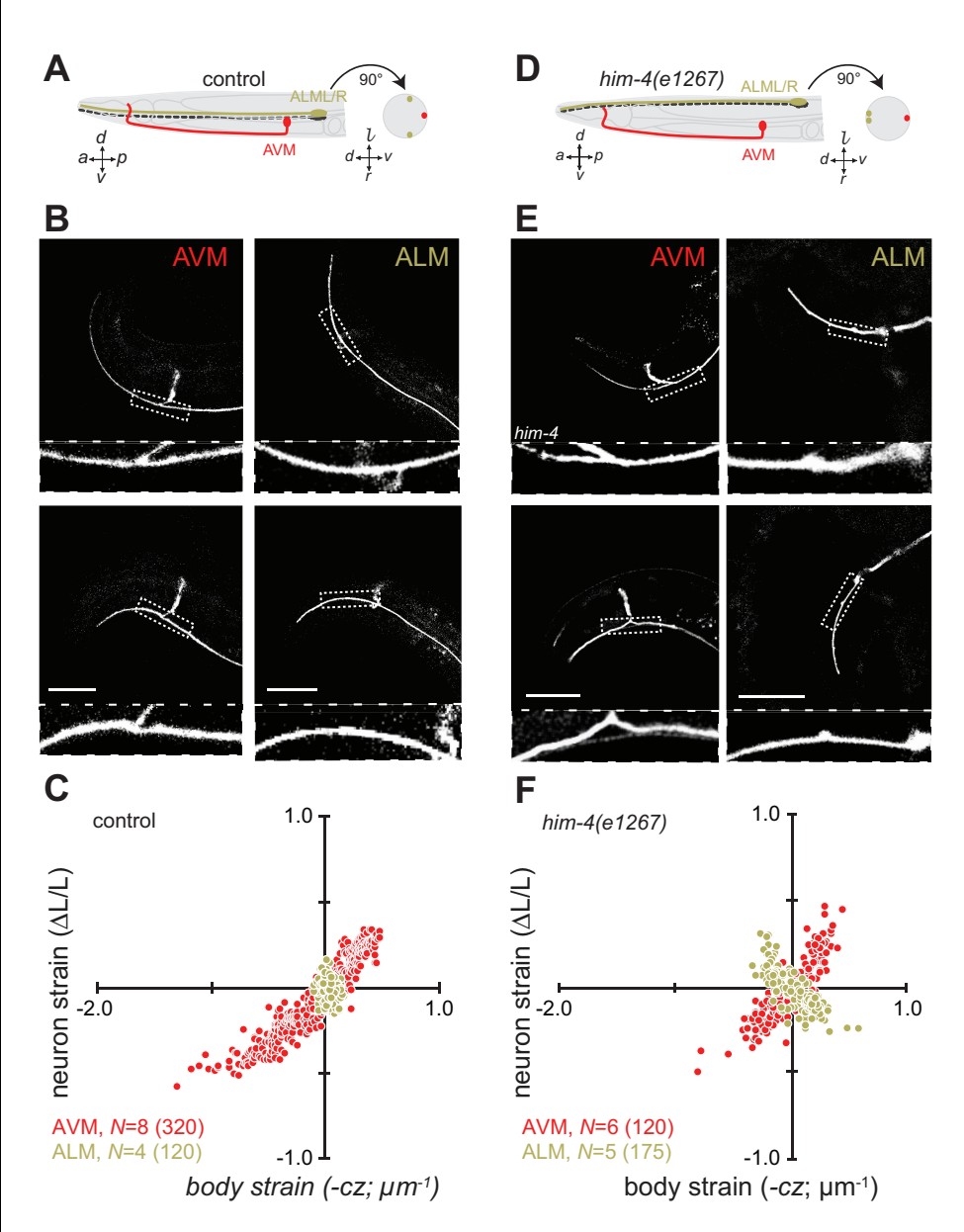

**Figure 2.** Axon position within tissues determines exposure to mechanical stress. (A) Schematic showing the position of AVM (red) and ALM (green) from the side (left) and in a cross-section (right) in control animals. (B) Still images of AVM and ALM during dorsal (top) and ventral bending (bottom) in control animals. Scale = 50 µm. Images are from *Video 7* (AVM) and *Video 8* (ALM). Anterior is to the left and ventral is down. (C) Normalized neuron length change ($\triangle L/L$, strain) is plotted against the axial component of the bending strain (curvature *times* distance from centerline, $c \cdot z$) during body bending. (D) Schematic showing the position of AVM (red) and ALM (green) from the side (left) and in a cross-section (right) in *him-4(e1267)* animals. The *him-4* mutation displaces ALM to the dorsal surface of the animal's body. (E) Still images of AVM and ALM during dorsal (top) and ventral bending (bottom) in *him-4(e1267)* animals. Scale = 50 µm. Images are from *Video 9* (AVM) and *Video 10* (ALM). (F) Normalized neuron length change ($\triangle L/L$, strain) plotted against the axial component of the bending strain (curvature *times* distance from centerline, $c \cdot z$) during body bending. AVM and ALM were visualized using the *uIs31* transgene driving GFP expression only in the TRNs. Sub-panels in (B) and (E) show a magnified view of the respective neuron taken from the area indicated by the dotted box. In panels (C) and (F), N = total number of axons analyzed per genotype and the number in parentheses gives the number of video frames analyzed. Animals that failed to move during the observation time were excluded from the analysis.

*Figure 2 continued on next page*

*Figure 2 continued*

The following figure supplement is available for figure 2:

**Figure supplement 1.** The length change (strain) of TRNs axons depends on body posture and anatomical position.

We next used the DER model to simulate TRN responses to mechanical stress as we varied the tension, bending stiffness, and twist rigidity around the initial values that we measured [tension (*Krieg et al., 2014*); stiffness (this study); or estimated (twist rigidity)]. The simulations revealed three shape classes for a neuron twisted around its longitudinal axis: straight axons, axons with localized helices, and plectonemes reminiscent of the ones created with in a twisted, elastic thread (*Figure 4A*, *Video 15*, *Video 3*). For a given shear modulus, the presence or absence of these shapes depended on the parameters of bending stiffness and tension for a given shear modulus (*Figure 4B*), axial stiffness (*Figure 4C*), and bending stiffness (*Figure 4D*). Similar to our experimental findings, simulated axons displayed multiple plectonemes at intervals of ~10–20 μm (*Figure 4E*, *Figure 4—figure supplement 3*). By exploring a larger parameter space, we found that plectonemes could also co-exist with less localized bends for a given shear modulus. The separation between these two phases depended on the values for tension and bending stiffness. These simulations predict that TRN axons will appear straight even when highly twisted, provided they have a sufficiently high bending stiffness and are held under axial stiffness. In this scenario, the generated torque is not strong enough to bend the neuron. Reducing either bending stiffness or end-to-end axial stiffness leads to the formation of localized helices for small values of twist and plectoneme-like supercoils at high levels of twist (see *Nizette and Goriely, 1999*).

A reduction of either bending stiffness or tension alone (in absence of torque or low shear modulus) was not sufficient to induce the formation of coils or plectonemes for 10-fold variations around the experimentally determined input moduli (*Figure 4B, C and D*). An increase in torque leads to the formation of localized, solitary plectonemes and further increases in torque induce multiple plectonemes. These results are consistent with analytical solutions (*Goriely and Tabor, 1998*) that specify a buckling stability criterion, $\tau_{crit}$ (*Equation 5*). The analytical expression for $\tau_{crit}$ underscores the interdependence of bending stiffness, $B$ (*Equation 1*), axial stiffness, $k$ (*Equation 2*), and twist rigidity (*Equation 3*).

$$\tau_{crit} = 2\frac{\sqrt{kB}}{C} \qquad (5)$$

For torques that exceed $\tau_{crit}$, the rod collapses into structural instabilities. Rearranging *Equation 5*, yields $k_{crit}$, which is inversely proportional to $B$:

$$k_{crit} = \frac{C^2\,\tau^2}{4B} \qquad (6)$$

This indicates the boundary (*Figure 4B,C and D*, green lines) separating areas that lack bends from those that contain bends and plectonemes. This boundary shifts to the left on each plot with lower $\tau$ and $C$.

In the classical description of rods, the mechanical parameters $B$, $C$, and $k$ all depend on Young's modulus, $E$, and the rod's cross-sectional area, $A$ (*Equations 1–4*). However, it is possible that bending stiffness, $B$, stretch stiffness, $k$, and twist rigidity, $C$, depart substantially from their predicted values. One source of variation is Poisson's ratio, which varies between 0 and 0.5 for

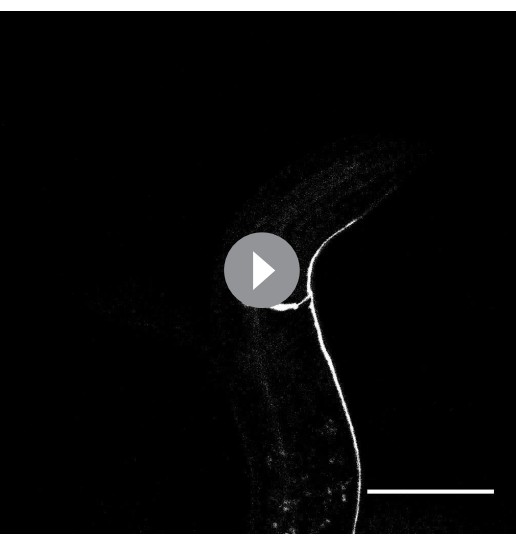

**Video 7.** Representative movie related to *Figure 2B* showing a GFP-labeled AVM neuron in a control animal. DOI: 10.7554/eLife.20172.014

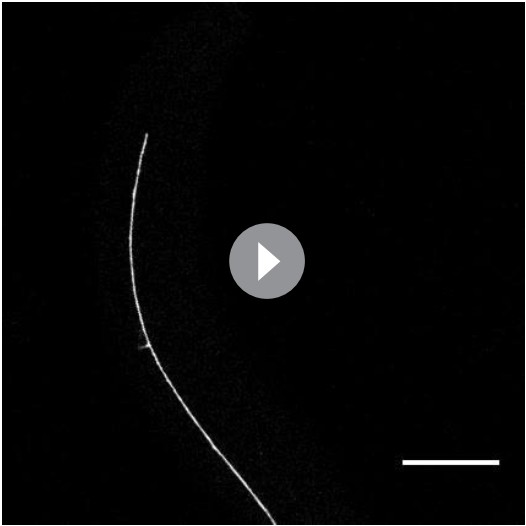

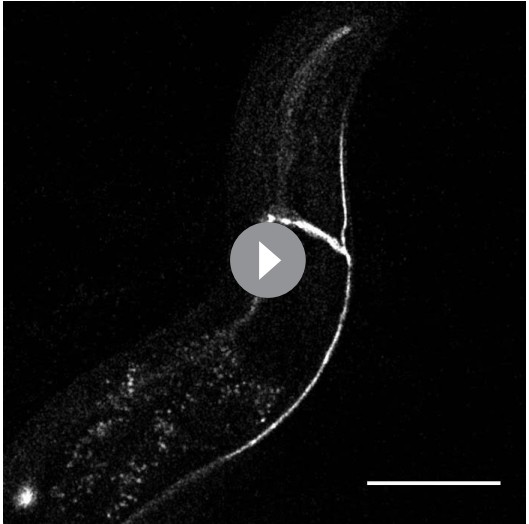

**Video 8.** Representative movie related to *Figure 2B* showing a GFP-labeled ALM neuron in a wild-type control animal.

**Video 9.** Representative movie related to *Figure 2E* showing a GFP-labeled AVM neuron in a *him-4(e1267)* mutant animal. Scale = 50 μm.

most materials and is known to deviate from this range for biological materials (e.g. *Reese et al., 2010*). Modeling the axon as an inhomogeneous material might also predict mechanical parameters that differ substantially from those resulting from our measurements. For instance, the bending stiffness estimated from the measured $E$ (*Figure 4—figure supplement 2*) using *Equation 1* yields $B = 1.5 \cdot 10^{-23}$ N m. However, estimates of $B$ based on the number ($N$) of MTs yields a bending stiffness of $1.26 \cdot 10^{-21}$ N m for a completely loose bundle ($B_{bundle} = N \cdot B_{MT}$) and $7.6 \cdot 10^{-20}$ N m in the limit of a completely crosslinked bundle ($B_{bundle} = N^2 \cdot B_{MT}$; *Guo et al., 2007*), with the assumption that the bending stiffness for a single MT $B_{MT} = 2.1 \cdot 10^{-23}$ N m (*Gittes et al., 1993*). This yields a 100-fold difference between the classical description (*Gittes et al., 1993*) and the microscopic model (*Bathe et al., 2008*; *Guo et al., 2007*). Similar arguments can be made for shear modulus and tension, justifying the large parameter space explored in our simulations (*Figure 4*).

In summary, these simulations provide a conceptual framework for analyzing the physical properties responsible for mechanical neuroprotection. In this model, neurons are stabilized against the effects of mechanical stress by high bending stiffness, end-to-end axial tension, or a combination of both factors. In particular, the simulations predict that the appearance of periodic loops is caused by an imbalance in axonal bending stiffness, axial tension, and torque. Furthermore, they suggest that shape distortions we observed in *mec-7(wy116);unc-70* and *ptl-1;unc-70* double mutants (*Figure 1*) and those reported in *mec-17* (*Topalidou et al., 2012*) and *mec-12(gm379)* (*Hsu et al., 2014*) might arise from a confluence of factors such as decreased bending stiffness, decreased axial tension, and/or an increase in the effective axonal shear modulus (torque).

## Spectrin forms a periodic, helical lattice in *C. elegans* axons and dendrites

To examine the architecture of actin-spectrin networks, we used super-resolution microscopy (stimulated emission depletion; STED) to visualize SPC-1 α-spectrin and UNC-70 β-spectrin in transgenic animals carrying fluorescent proteins fused to their C- and N-termini, respectively (*Figure 5—figure supplement 1*). In agreement with studies of actin-spectrin networks in mammalian and invertebrate axons and dendrites (*D'Este et al., 2016*, *2015*; *He et al., 2016*; *Pielage et al., 2008*; *Xu et al., 2013*; *Zhong et al., 2014*), α– and β-spectrin formed periodic structures in wild-type *C. elegans* TRNs in vitro (*Figure 5*, *Video 16*) with an average spacing of 197 ± 13 nm and 199 ± 10 nm, respectively (p=0.64, Mann-Whitney test), suggesting that the spectrin tetramer resides at its fully extended contour length in these cells.

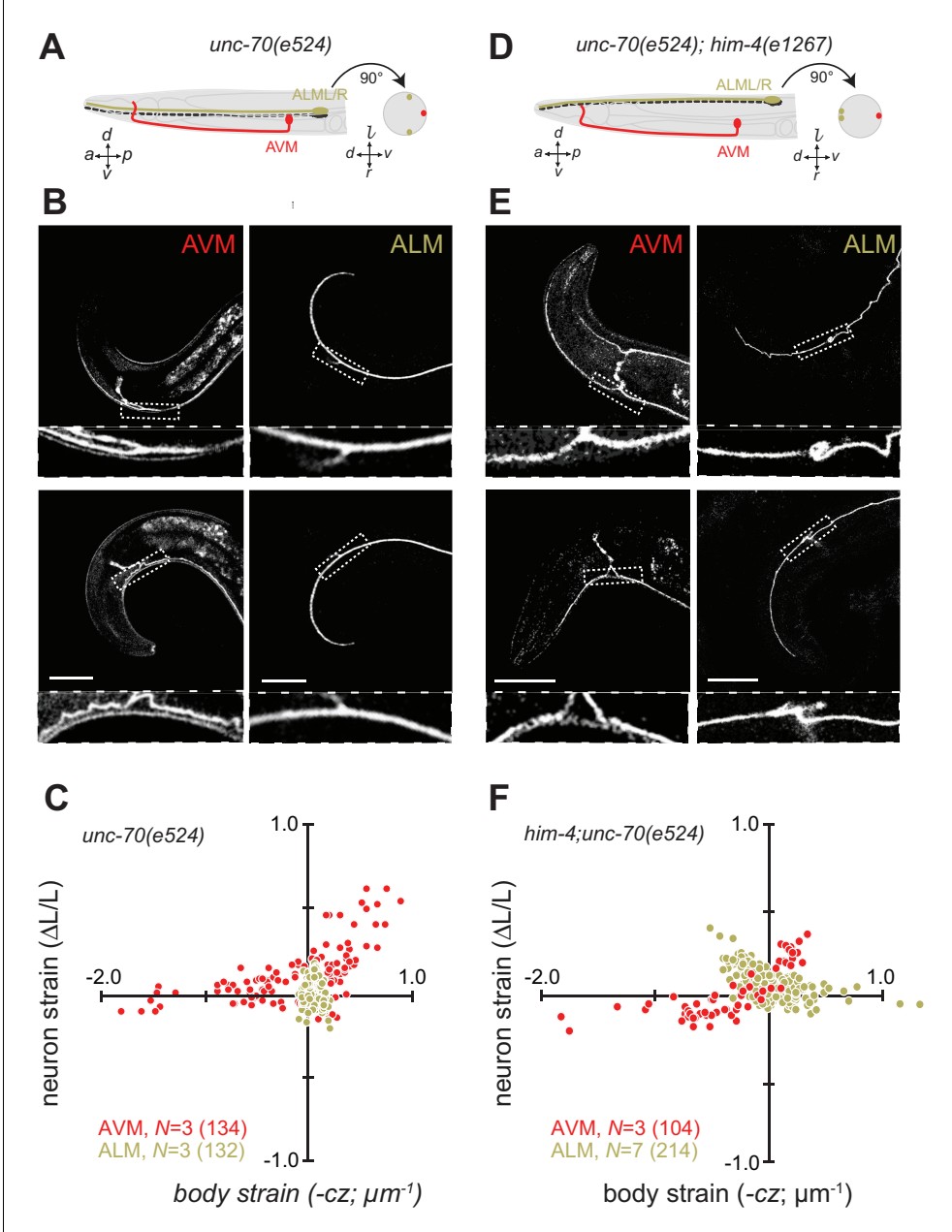

**Figure 3.** Axons position with tissue determines sensitivity to defects in *unc-70 β*-spectrin. (**A**) Schematic showing the position of AVM (red) and ALM (green) from the side (left) and in a cross-section (right) in *unc-70(e524)* missense mutants. (**B**) Still images of AVM and ALM during dorsal (top) and ventral bending (bottom) in *unc-70(e524)* animals. Scale = 50 µm. Images are from ***Video 11*** (AVM) and ***Video 12*** (ALM). Anterior is to the left and ventral is down. (**C**) Normalized neuron length change ($\triangle L/L$, strain) plotted against the axial component of the bending strain (curvature $\times$ distance from centerline, $c \cdot z$) during body bending. (**D**) Schematic showing the position of AVM (red) and ALM (green) from the side (left) and in a cross-section (right) in *unc-70(e524); him-4(e1267)* animals. The *him-4* mutation displaces ALM to the dorsal surface of the animal's body. (**E**) Still images of AVM and ALM during dorsal (top) and ventral bending (bottom) in *him-4(e1267)* animals. Scale = 50 µm. Images are from ***Video 13*** (AVM) and ***Video 14*** (ALM). (**F**) Normalized neuron length change ($\triangle L/L$, strain) is plotted against the axial component of the bending strain (curvature *times* distance from centerline, $c \cdot z$) during body bending. AVM and ALM were visualized using the *uIs31* transgene driving GFP expression only in the TRNs. Sub-panels in (**B**) and (**E**) show a magnified view of the respective neuron taken from the area indicated by the dotted box. In panels (**C**) and (**F**), *N* = total number of axons analyzed per genotype and the number in

*Figure 3 continued on next page*

Krieg *et al*. eLife 2017;6:e20172. DOI: 10.7554/eLife.20172

*Figure 3 continued*

parentheses gives the number of video frames analyzed. Animals that failed to move during the observation time were excluded from the analysis.

The following figure supplement is available for figure 3:

**Figure supplement 1.** The length change (strain) of TRNs axons depends on body posture and anatomical position.

---

Periodic spectrin structures observed in GFP::SPC-1 α-spectrin transgenic neurons in vitro were eliminated upon addition of the actin depolymerizing agent Latrunculin A (*Figure 5c*) and by loss of UNC-70 β-spectrin (*Figure 5D*). Interestingly, α-spectrin remained organized into periodic structures in *unc-70(e524)* mutants (*Figure 5E*), but displayed a slightly shorter period of 188 ± 13 nm (p=0.003, Mann-Whitney U-test; *Figure 5F*). The 5% shorter period in *unc-70(e524)* neurons is consistent with the decreased axial tension found in these mutants in vivo (*Krieg et al., 2015*). Loss of *ptl-1* expression, by contrast, had no obvious effect on the distribution of α-spectrin (*Figure 5—figure supplement 2)*, which still formed periodic structures with a period similar to that observed in wild-type neurons: 195 ± 10 nm (n = 10, U-test, p=0.42). Taken together, these observations indicate spectrin forms a periodic structure sensitive to axial tension, but not defects in the microtubule cytoskeleton.

## Microtubule bundles are disrupted in *mec-7(wy116)* β-tubulin and *ptl-1(ok621)* tau mutants

A bundle of long, 15-protofilament MTs overlap to fill the TRN cytoplasm (*Chalfie and Thomson, 1982*). We used serial-section transmission electron microscopy (ssTEM) to determine how genetic defects in the microtubule and spectrin cytoskeleton affect these MTs and their organization within the bundle. The distribution of edge-to-edge distances between adjacent microtubules are well described by a log-normal distribution with a geometric mean of 9.8 nm for wild-type TRNs (*Figure 6Aii*). The MTs are organized into an approximately hexagonal bundle, as demonstrated by 60°, 120°, and 180° being the dominant angles between three adjacent MTs (*Figure 6Aiii*, *Figure 6—figure supplement 1*). In *mec-7(wy116)* TRNs, by contrast, the distribution of edge-to-edge distances was altered such that the geometric mean decreased to 4 nm, or about half the value in wild-

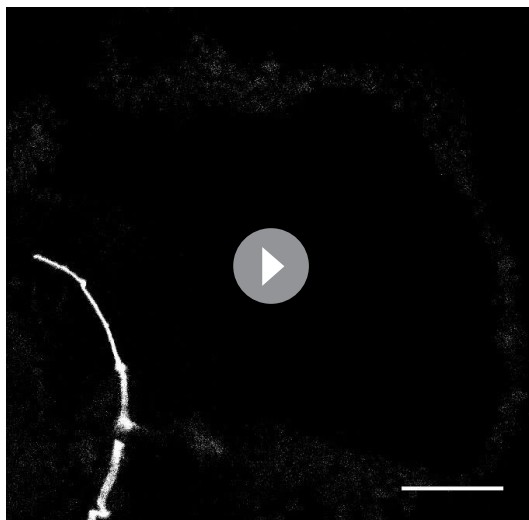

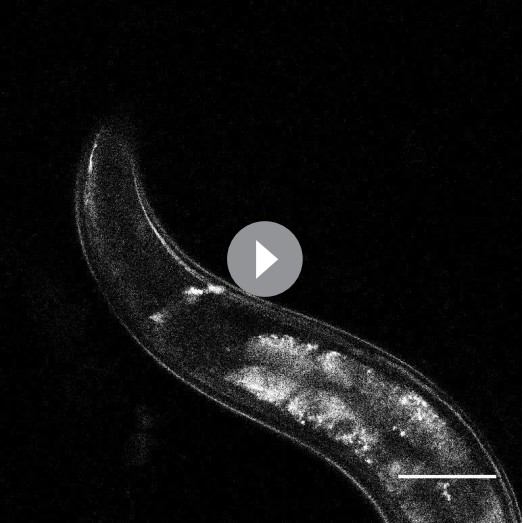

**Video 10.** Representative movie related to *Figure 2E* showing a GFP-labeled ALM neuron in a *him-4(e1267)* mutant animal. Scale = 50 µm.

**Video 11.** Representative movie related to *Figure 3B* showing a GFP-labeled AVM neuron in a *unc-70(e524); uIs31* animal.

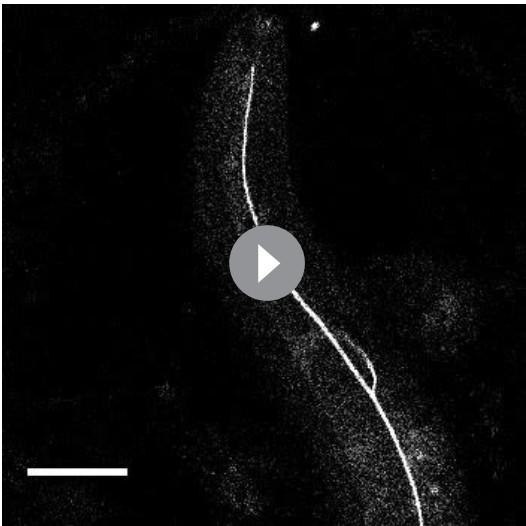

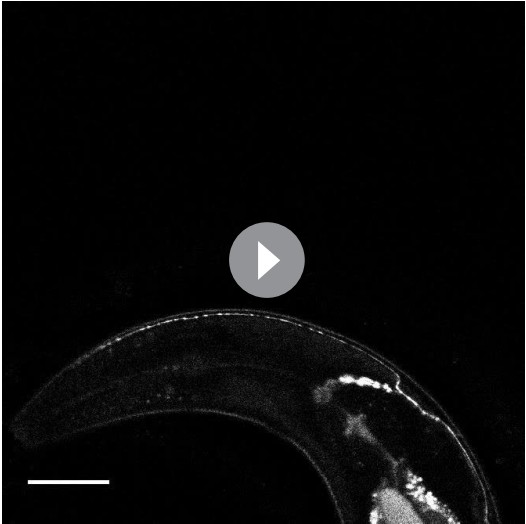

**Video 12.** Representative movie related to *Figure 3E* showing a GFP-labeled ALM neuron in a *unc-70(e524); uIs31* animal.

**Video 13.** Representative movie related to *Figure 3* showing a GFP-labeled AVM neuron in a *him-4(e1267; unc-70(e524);uIs31* animal. Scale = 50 μm.

type (*Figure 6Bii*). The variance in this distribution was nearly three-fold that found in wild-type ($\sigma^2$ = 32 and 89 nm$^2$ in wild-type and *mec-7(wy116)* respectively). Despite this, a hexagonal arrangement of MTs was also detected in *mec-7(wy116)* axons (*Figure 6Biii*) with similar distribution of inter-MT angles to wild type (p=0.2; Watson U$^2$ for radially distributed data), although the 120° and 180° peaks seemed collapsed together in the histograms. However, individual MTs often separated from the main bundle and adjacent MTs often contacted one another in these mutants (*Figure 6Bi*), situations that we have seldom if ever observed in wild-type TRNs (*Bellanger et al., 2012*; *Cueva et al., 2012*, *2007*, *2007*; *Hueston et al., 2008*; *Richardson et al., 2014*). Qualitatively similar structural phenotypes were found in some but not all samples taken from *mec-7(wy116); unc-70(e524)* double

mutants (*Figure 6C*). This variable phenotype likely accounts for the bimodal distribution of edge-to-edge microtubule distances that we observed (*Figure 6Cii*).

Similar to *mec-7(wy116)* missense mutants, MTs were closer together in *ptl-1(ok621)* null mutants than in wild-type axons (*Figure 6Di, Dii*). The distribution of edge-to-edge distances had a geometric mean of 6.7 nm and a variance of 81 nm$^2$. Meanwhile, the hexagonal geometry of MT cross-sections was largely unaffected (*Figure 6Diii*) despite our observation that adjacent MTs often collapsed onto one another (*Figure 6Di*). In *ptl-1(ok621);unc-70(e524)* double mutants (*Figure 6E*), the distribution of edge-to-edge distances was similar to that observed in the *ptl-1(ok621)* single mutant (*Figure 6Eii*): the geometric mean was 6.9 nm and its variance was 114 nm$^2$.

This analysis also revealed that *mec-7 β*-tubulin and *ptl-1* tau mutants had fewer and shorter microtubules than wild type (*Figure 6F and G*; *Figure 6—figure supplement 2*). Given that

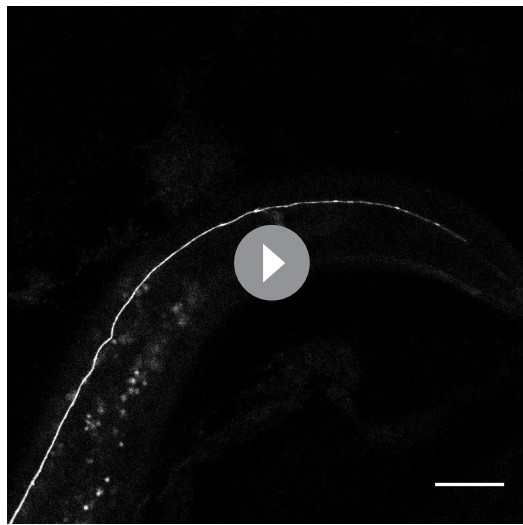

**Video 14.** Representative movie related to *Figure 3* showing a GFP-labeled ALM neuron in a *him-4(e1267); unc-70(e524);uIs31* animal. Scale = 50 μm.

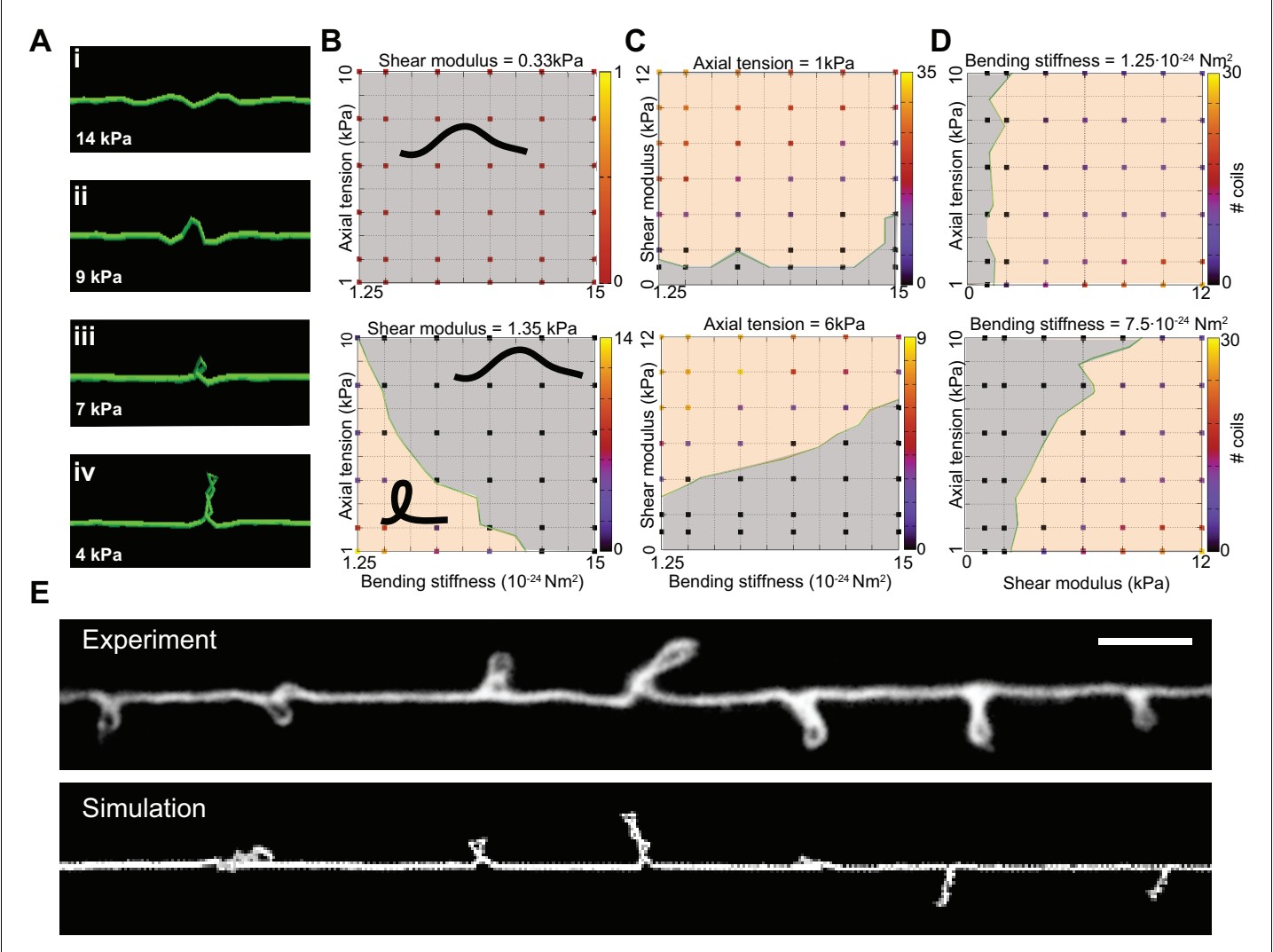

**Figure 4.** Simulation of axon shape with varying mechanical properties. (A) Representative images derived from simulations based on a discrete elastic rod or DER model (Methods, Appendix) using the following parameters: $B = 5\cdot10^{-24}$ N m$^2$, $G = 1.35$ kPa and the value of for tension as indicated in each panel. In this example, tension decreases from top to bottom: (i) Stiff and twisted neuron remains straight. (ii) A slight decrease in tension causes local bends and spiraling. (iii) Further reduction in tension leads to the transfer of torsional energy into bending of the rod and the formation of local coils. (iv) Further reduction in tension causes the coils in the model neuron to rotate, leading to plectonemes spaced at regular intervals. (B) Parameter space plot of a simulated neuron subjected to low torque (top, shear modulus, $G = 0.33$ kPa) or higher torque (bottom $G = 1.35$ kPa) and varying axial stiffness and bending stiffness. For panels (B–D), vertex color shows the number of coils predicted for pair of values for tensile modulus and bending rigidity, according to the scale to the right of each plot. Gray areas indicate conditions with no coils; beige areas indicate conditions with at least one coil; green line indicates the boundary between the two shape phases. (C) Parameter space plot of a simulated neuron under low (top, 1 kPa) and high tension (bottom, 6 kPa) and varying shear modulus and bending stiffness. (D) Parameter space plot of a simulated neuron with low (top, $1.25\cdot10^{-24}$ Nm$^2$) and high bending stiffness (bottom, $7.5\cdot10^{-24}$ Nm$^2$) and varying shear and elastic moduli. Bending stiffness was calculated according to $B=EI$, with $I$ as the second moment of inertia. (E) Comparison of the morphology of an TRN in *ptl-1(ok621); unc-70(e524)* double mutant animals with a simulation result of our DER model with the following parameters: bending stiffness = $7.5\cdot10^{-24}$ Nm$^2$; shear modulus = 8 kPa; axial stiffness = 1 kPa. Scale = 10 μm.

The following figure supplements are available for figure 4:

**Figure supplement 1.** Atomic force microscopy on axons of isolated TRNs.

**Figure supplement 2.** Elastic moduli of axons of touch receptor neurons in vitro.

**Figure supplement 3.** The distance between plectonemes in mutant and simulated axons.

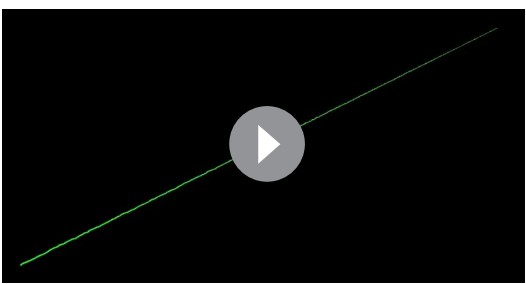

**Video 15.** Numerical simulation of an Euler Kirchhoff rod using the discrete elastic rod model related to *Figure 4* (see *Figure 4* and Materials and methods) with tension $k = 2$ μN/m, bending stiffness $B = 5 \cdot 10^{-24}$ N m$^2$, and shear modulus $G = 1.35$ kPa. Friction with the surrounding tissue is modeled as a Stoke's drag (dynamic viscosity = 1 PI). A twist of 360° per 1.5 μm is evenly distributed along the rod similar to the one observed in some *ptl-1(ok621)* electron micrographs (*Figure 6d*). Together with clamped boundary conditions this results in the dynamic formation of helical structures that collapse to plectonemes.

bundle stiffness is expected to be proportional to the number of filaments in the bundle (*Bathe et al., 2008*; *Guo et al., 2007*), this finding suggests that bending stiffness is decreased in both of these mutants. This is not the only structural feature that suggests a decrease in bending stiffness: The fact that neighboring MTs contact one another in mutant but not wild-type axons suggests that mutants lack a spacer or linker molecule leading to a reduction in its cross-sectional area. A decrease in area and in the number of cross-links is expected to decrease the bending stiffness of the bundle (*Bathe et al., 2008*; *Tolomeo and Holley, 1997*). Thus, we propose that defects that reduce MT length, number, and inter-microtubule spacing also decrease the axon's bending stiffness.

Not all MT profiles in *ptl-1* and *ptl-1;unc-70* axons conformed to those expected from the cross-section of a cylindrical MT. Some cross-sections appeared to be oblique or nearly longitudinal (*Figure 6—figure supplement 3A*). Rarely, we captured TRN profiles in *mec-7(wy116);unc-70* mutants consistent with having sectioned a local coil within the TRN axon (*Figure 6—figure supplement 3B–D*). One of the most striking and consistent observations in the ssTEM datasets from *ptl-1* and *ptl-1;unc-70* mutants was that outer MTs rotated and tilted away from the central axis of the bundle (*Video 17*) in manner that generated a twist of the entire MT bundle. We applied particle image velocimetry (PIV) techniques to adjacent thin sections to quantifying this structural phenomenon (*Figure 7A, B and C*). This approach yields an estimate of the pitch of the helical twist in the entire bundle and showed that the pitch angle was small near the center and increased linearly with across the bundle's radius (*Figure 7D*). Notably, we also observed helical perversions in twisted bundles, in which the MT bundle reverses between clockwise and counter-clockwise twists (*Figure 7D*, *Video 17*). Thus, *ptl-1* loss of function resulted in fewer, shorter MTs that were organized into a twisted bundle, findings consistent with the idea that *ptl-1* mutant axons have both decreased bending stiffness and increased torque.

## *C. elegans* tau protein binds MTs in vivo and contributes to mechanical neuroprotection

Based on our observation that *mec-7(wy116)* and *ptl-1(ok621)* mutants exhibit similar defects in MT bundle structure, we hypothesized that *wy116* disrupted the association between the PTL-1 protein and microtubules. To test this idea in living animals, we used CRISPR/Cas9-based genome editing (*Dickinson et al., 2015*) to create a PTL-1::mNeonGreen (PTL-1::mNG) fusion protein and performed a FRAP assay in wild-type and *mec-7* β-tubulin mutants (*Figure 8*). In agreement with prior work (*Konzack et al., 2007*) and consistent with the idea that PTL-1 protein diffuses along microtubules (*Hinrichs et al., 2012*), we observed rapid but incomplete recovery of PTL-1::mNG fluorescence in control axons. On average, the apparent diffusion constant for PTL-1 in living axons was $0.12 \pm 0.1$ μm$^2$ s$^{-1}$ (mean ± s.d., $n= 47$), which agrees with values reported for mammalian tau in vitro ($0.17 \pm 0.03$ μm$^2$ s$^{-1}$; *Hinrichs et al., 2012*). Both the recovery rate and the mobile fraction increased in *mec-7(ok2152)* β-tubulin null mutants and in *mec-7(wy116)* β-tubulin missense mutants. By comparison, PTL-1 mobility was unaffected in *unc-70(e524)* mutants (*Figure 8*). In principle, the increased PTL-1 mobility found in *mec-7(wy116)* could reflect a decrease in the number of MTs (*Figure 6*), a decrease in the affinity of PTL-1 for mutant MTs, or a combination of both factors. We favor the idea that the *wy116* allele affects the affinity for PTL-1 since *unc-70(e524)* had no detectable effect on PTL-1 mobility (*Figure 8D and E*) despite having fewer MTs (*Table 2*). Together, these

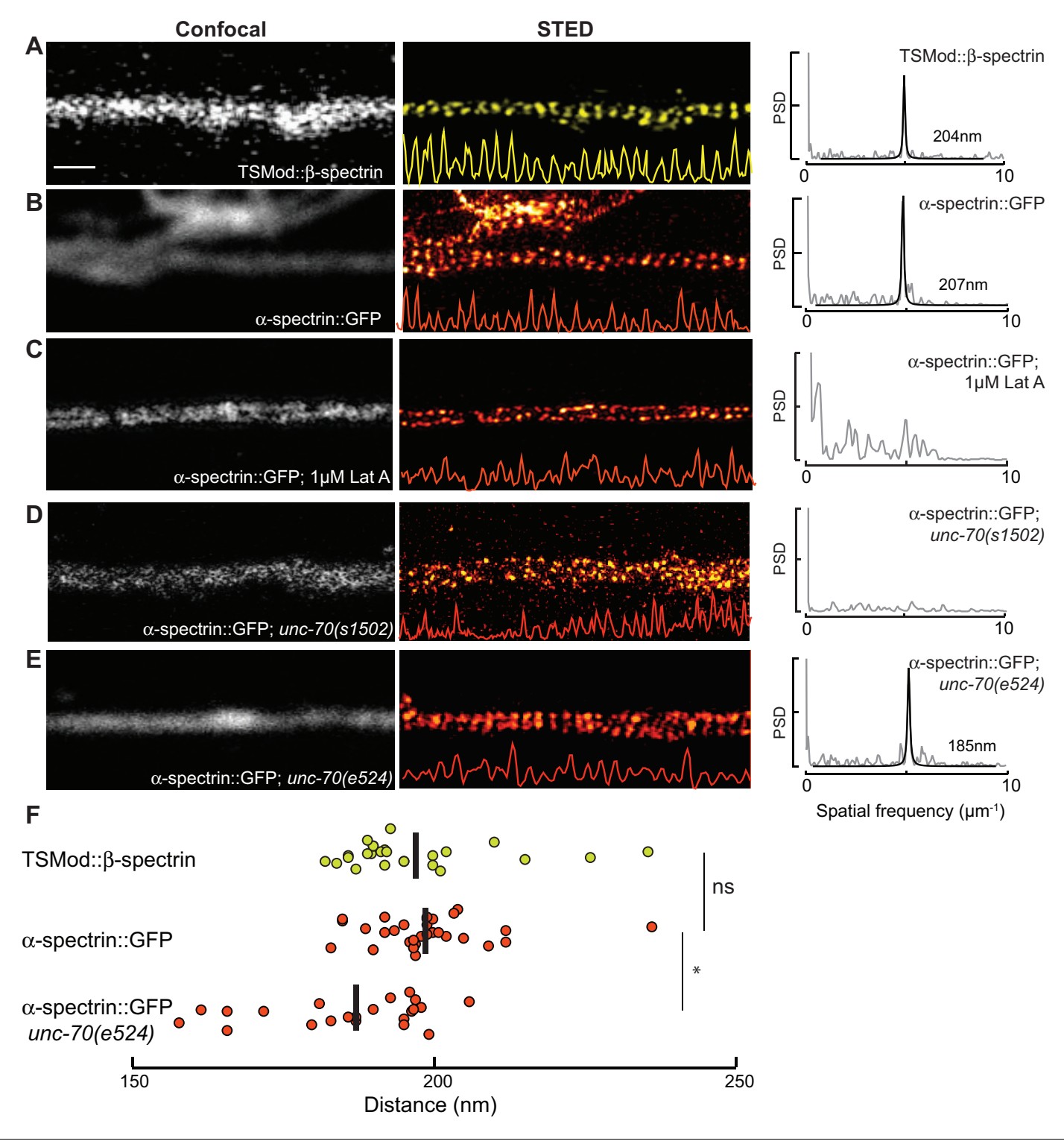

**Figure 5.** Spectrin forms an actin-dependent, periodic cytoskeleton in vitro. (A–E) Representative confocal (left), STED super-resolution images (center), and power spectral density (PSD) profiles (right) of TSMod::$\beta$-spectrin (A) and $\alpha$-spectrin SPC-1::GFP (B–E) in *C. elegans* axons in vitro. In panels (A–E), yellow ($\beta$-spectrin) and red ($\alpha$-spectrin) traces show intensity profiles derived from STED images. Scale bar = 1 μm. A single PSD curve (gray) and Lorentzian fit (black) is shown for each STED image. Cells were dissociated from transgenic embryos and cultured on glass coverslips for three days. Similar images were obtained from at least 20 axons examined per condition (genotype, label, treatment) from at least three biological replicates. (A, B) Spectrin in control axons (left, center) and PSD showing strong peaks at 204 and 207 nm for TSMod::$\beta$-spectrin and $\alpha$-spectrin::GFP, respectively. (C)

*Figure 5 continued on next page*

*Figure 5 continued*

SPC-1::GFP in control neurons treated with the actin depolymerizing agent, Latrunculin A (1 μM) (left, center). The PSD lacks any strong peaks, indicating that this treatment disrupts spectrin periodicity. (D) SPC-1::GFP in *unc-70(s1502)* β-spectrin null mutant (left, center). As in panel C, the PSD (right) lacks any clear peaks. (E) SPC-1::GFP in *unc-70(e524)* β-spectrin missense mutant. The PSD has a single strong peak at 185 nm. (F) Average distance between adjacent spectrin labels for TSMod::β-spectrin (top, n = 27) in control neurons; α-spectrin::GFP (middle, n = 31) in control neurons; and SPC-1::GFP (bottom, n = 20) in *unc-70(e524)*. Spacing computed as the inverse distance of the dominant peak in PSD.

The following figure supplements are available for figure 5:

**Figure supplement 1.** Fluorescent labeling strategy on α/β-spectrin.

**Figure supplement 2.** Spectrin network periodicity is similar in control and *ptl-1(ok621)* TRNs.

results support the idea that PTL-1 binds to MTs in vivo and that the T409I mutation in MEC-7 β tubulin interferes with PTL-1 binding to MTs.

## The spectrin network stabilizes TRN microtubules against fracture

To address the question of how loss or disruption of the periodic spectrin network affects MTs, we compared MTs in *unc-70(e524)* mutants, which retain a periodic spectrin cytoskeleton, and *unc-70 (s1502)* mutants, in which the spectrin network is disrupted (*Figure 9A,B and C*). Given the differential sensitivity of AVM and ALM neurons to mechanical stress in *unc-70* mutants (*Figure 3*), we examined MTs in both cell types. In accord with prior studies of wild-type animals, we found that MTs were shorter in AVM neurons than in the ALM neurons, on average (*Figure 9D*; *Figure 9—figure supplement 1*; *Table 2*). Thus, mechanical stress may be a factor that limits MT length in wild-type axons, since AVM neurons undergo more extreme cycles of compression and extension than do the ALM neurons (*Figure 2*).

Neither the *e524* partial loss-of-function nor the *s1502* null allele of *unc-70* caused a further reduction in MT length or number in AVM neurons, but both alleles were associated with a decrease in MT length in the ALM neurons (*Figure 9D and E*). This differential sensitivity to *unc-70* mutation likely arises from the fact that MTs are already shorter in AVM and do not shorten further. Moreover, MT length in AVM is similar to the buckling wavelength observed under compression in *unc-70* mutants (*Figure 9—figure supplement 2*). The variance in the *number* of MTs was higher in *unc-70(s1502)* mutant ALM neurons than that in either wild-type (F-test p=0.01) or *unc-70(e524)* (F-test p=0.05; *Figure 9F*. *Figure 9—figure supplement 1*), but no difference in average values was detected (*Figure 9E*). Related to the expected mechanical stresses generated in these animals, *e524* animals have a loopy and uncoordinated locomotion phenotype and *s1502* animals are paralyzed (*Hammarlund et al., 2000*). As a result, *e524* animals are expected to generate more intense movement-induced mechanical stresses than *s1502* animals. In light of these differences, we interpret the increased variance in MT as an indication that some long MTs persist in *s1502* mutants because mechanical stresses are decreased in this genetic background and other MTs are shortened due to the loss of actin-spectrin networks.

To further examine the possibility that the shorter MTs in the β-spectrin mutants might result from mechanically enhanced fracture and/or

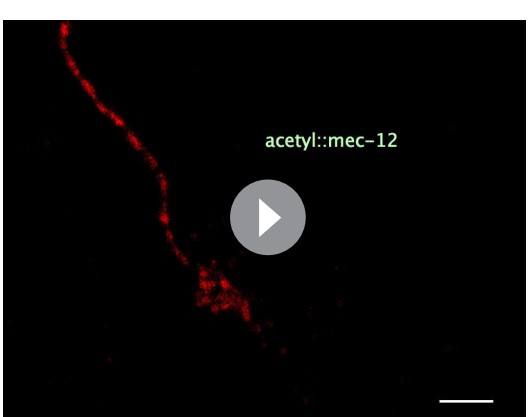

**Video 16.** Animation related to *Figure 5* showing an overlay of confocal and STED images showing a TRN identified by anti-acetylated tubulin immunofluorescence together with a cluster of unidentified neurons.

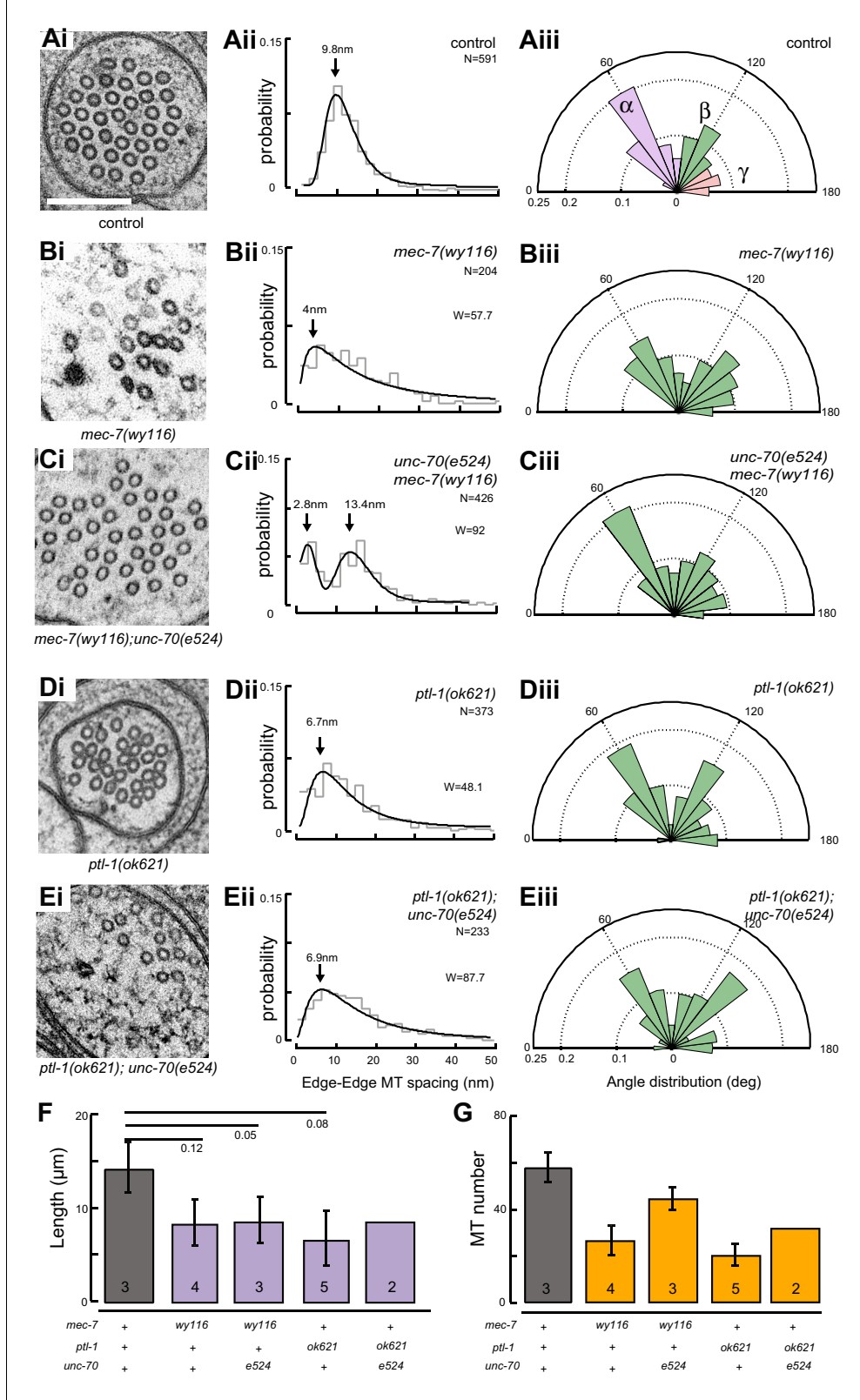

**Figure 6.** Loss of PTL-1 tau and defects in MEC-7 β tubulin disrupt the organization of microtubule bundles in vivo. (A–E) Organization of microtubule bundles in control and mutant TRNs illustrated by electron micrographs (sub-panel i), the edge-to-edge distances between adjacent MTs (sub-panel ii), and inter-MT angles (sub-panel iii). The genotypes shown are. (A) Control: *uIs31[mec-17p::GFP]*. (B) *mec-7(wy116); uIs31*. (C) *unc-70(e524);mec-7(wy116); uIs31*. (D) *ptl-1(ok621); uIs31*. (E) *ptl-1(ok621);unc-70(e524); uIs31*. For electron micrographs (sub-panels i), the scale bar is 200 nm. Between 13 and 25

*Figure 6 continued on next page*

*Figure 6 continued*

sections drawn from at least five datasets/genotype were analyzed to derive the data in sub-panels ii and iii. Inter-MT angles were computed as described in the Materials and methods and illustrated in *Figure 6—figure supplement 1*. The variances in the distribution of edge-to-edge MT spacing are significantly different if $W$ is larger than the critical value $W_c$ = 3.85 (for p>0.05), using Levene's test adjusted for the median. The distribution of angles in mutants was not significantly different from wild type according to Watson $U^2$ test for circular distributed data. (**F**) Average microtubule length as a function of the indicated genotypes. Error bars indicate S.E.M. for the number of datasets indicated on each bar. (**G**) Average microtubule number for the indicated genotypes. Error bars indicate S.E.M. for the number of datasets indicated on each bar.

The following figure supplements are available for figure 6:

**Figure supplement 1.** Method for measuring the angles between three neighboring MTs.

**Figure supplement 2.** Microtubule number as a function of distance for (**A**) control and (**B**) *mec-7(wy116)* single and *mec-7(wy116); unc-70(e524)* double mutant and (**C**) for *ptl1(ok621)* and *ptl-1(ok621); unc-70(e524)* double mutants.

**Figure supplement 3.** Defects in MT bundle architecture and examples of putative plectoneme structures.

disassembly, we generated *unc-70(e542);oxIs95[epi::UNC-70]* and *unc-70(s1502);oxIs95[epi::UNC-70]* animals in which wild-type UNC-70 $\beta$-spectrin was expressed in the epidermal cells that engulf the ALM neurons. We found that MT length and number in *unc-70(e542);oxIs95[epi::UNC-70]* transgenic animals were similar to those of *unc-70(e542)* mutants (*Table 2*; *Figure 9D and E*). However, the epi:: UNC-70 transgene appeared to restore MT length to nearly wild-type levels in *unc-70(s1502)* mutants (*Figure 9D*). One plausible explanation for this result is that the mutant $\beta$-spectrin present in *unc-70 (e524)* animals acts in a dominant fashion and interferes with the mechanical stabilization that would otherwise result from the expression of wild-type UNC-70 $\beta$-spectrin in the epidermis.

Together, these findings suggest that the longer ALM MTs are more susceptible to bending-induced fracture than the shorter AVM MTs. In addition, an analysis of MT length and number in $\beta$-spectrin mutants suggests that the actin-spectrin network provides a lateral support that protects the MT bundle against deformation via both cell-autonomous and non-autonomous mechanical stabilization.

## The MT bundle can account for the torque build-up in *ptl-1;unc-70* mutants

Experimental measurement and simulations support the hypothesis that supercoils and plectonemes are revealed when axons experience a combination of high torque, low bending stiffness, and reduced axial stiffness (*Figure 4*). The discovery of twisted MT bundles in *ptl-1* single mutants and *ptl-1;unc-70* double mutants (*Figures 6* and *7*) points toward the MT bundle as the source of torque. To investigate this possibility, we created a *ptl-1;unc-70;mec-7(e1434)* triple mutant. *e1434* is a temperature-sensitive, semi-dominant, missense allele that encodes a P171I mutation in MEC-7 $\beta$-tubulin (*Savage et al., 1989*). This allele is linked to a reduction in MT number and length in animals grown at restrictive temperatures but is not known to display any clear shape defects (*Chalfie and Thomson, 1982*). Control experiments revealed that shape defects in the TRNs of *ptl-1;unc-70* double mutants were more pronounced for growth at 20°C than at 15°C (*Figure 10A*). When *ptl-1;unc-70;mec-7(e1434)* triple mutants were cultivated at a permissive temperature (15°C) that allows for nearly wild-type MT assembly (*Chalfie and Thomson, 1982*), the TRN axons contained loops, twists, and plectonemes. Strikingly, axons were smooth in the triple mutant following growth at 20°C, where MT assembly is defective (*Figure 10B*). In summary, these findings indicate that the formation of loops, twists, and plectonemes in mutant axons depends on intact microtubules, likely due to the induction of torque resulting from their assembly into twisted bundles.

## Discussion

In this study, we combined experiment and theory to reveal how axial tension and bending stiffness balance torque to protect long, thin TRN axons from mechanical deformation. The picture emerging from this work is that actin-spectrin networks provide tension, MT bundles confer stiffness, and

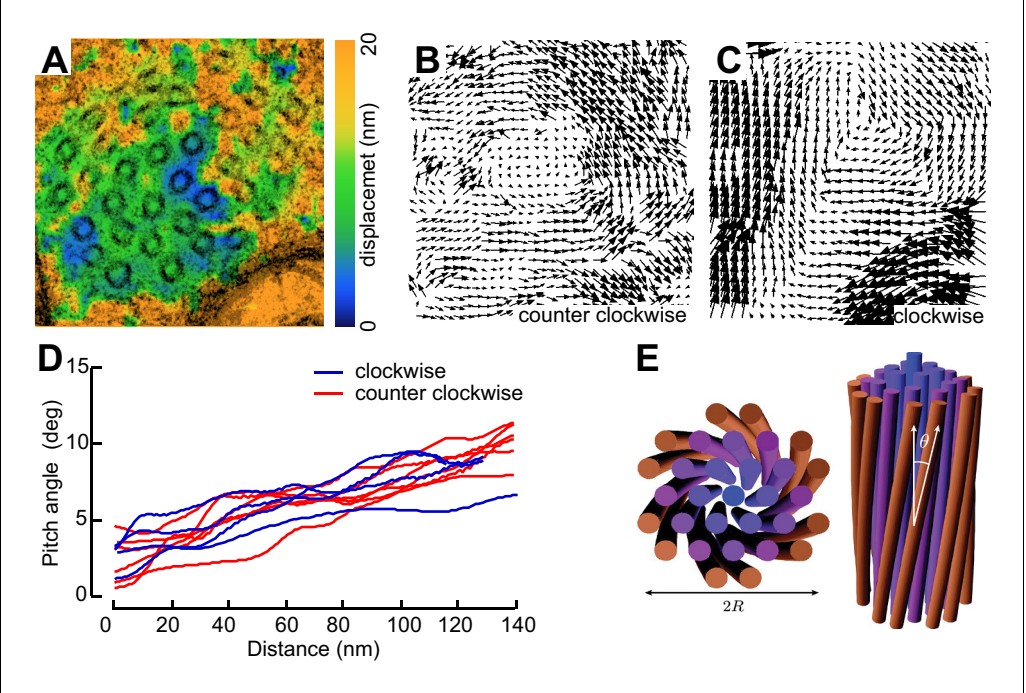

**Figure 7.** MTs twist around one another in PTL-1 tau mutants. (**A**) Minimum intensity projection (MIP) of two consecutive electron micrographs of an *unc-70;ptl-1* double mutant TRN superimposed on a heat map showing the magnitude of the displacement vector derived from PIV analysis of this image pair. Color scale indicates displacement magnitude. See also *Video 17*. (**B**) Flow field for the pair of sections shown in **A**. The size and direction of the arrows indicate displacement of the optical density between these frames. Note the counterclockwise rotation of the bundle around the central axis. (**C**) Flow field for a different pair of consecutive sections, showing clockwise rotation, in agreement with the helical perversion seen in *Video 17*. (**D**) Plot of the helical pitch angle as a function of the radius from the center of the bundle. (**E**) Schematics of the helical pitch angle of a twisted filament bundle.
Reproduced with permission from Figure 1 from *Grason and Bruinsma (2007)* (© copyright American Physical Society, 2007. All Rights Reserved).

microtubule-associate proteins like tau minimize torque within MT bundles. Together, these factors combine to protect axons from mechanical damage. Simulations based on Euler-Kirchhoff theory revealed that the balance of tension and torque stabilizes axons and dendrites and show that plectonemes result when torque overcomes tension (*e.g. Figure 4A.*). Experimental observations show that UNC-70 $\beta$-spectrin forms an extended cytoskeleton similar to that in mammalian neurons (this study and *D'Este et al., 2016*; *Xu et al., 2013*) and is essential for maintaining proper axial tension (*Krieg et al., 2014*). Changes in tension have also been linked to pearling, a phenomenon in which tubes under tension form bulges that resemble pearls on a string (*Bar-Ziv et al., 1999*; *Pullarkat et al., 2006*). Although pearling is driven by a competition between membrane tension and actin rigidity (*Bar-Ziv et al., 1999*), the periodic defects we found in tau-spectrin double mutants occur due to an unbalanced torque-tension coupling.

The shape defects seen in tau-spectrin double mutant axons were associated with dramatic rearrangements of the MT bundle. Whereas wild-type axons contain a regularly spaced array of MTs, axons lacking PTL-1 tau have MTs that contact one another and spiral around each other. How might PTL-1 tau oppose contact between adjacent MTs? Like other tau proteins, PTL-1 has a MT-binding domain in its C-terminus and an intrinsically disordered N-terminal projection domain harboring dozens of glutamate/proline repeats (*Chung et al., 2015*). These repeats have been proposed to function as a polyelectrolyte brush (*Wegmann et al., 2013*), to regulate the spacing between MTs in bundles and confer steric stabilization against aggregation (*Chung et al., 2015*; *Goedert et al., 1996*). Our data are likewise consistent with the idea that PTL-1 tau and related

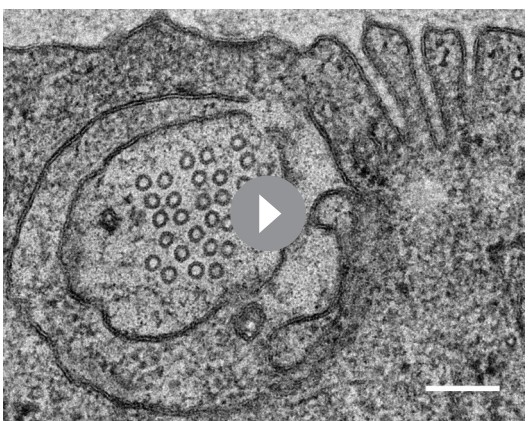

**Video 17.** Image stack derived from an ssTEM dataset of a *ptl-1(ok621); unc-70(e524)* mutant ALM neuron, related to *Figure 7A and C*. Initially, microtubules twist around one another in a counterclockwise manner and subsequently they twist in the opposite direction, a phenomenon known has helical perversion. Scale = 100 nm.

MAP2 proteins enable MTs to glide past one another during mechanical stress-induced compression and elongation. We infer that PTL-1 allows sliding of MTs in a bundle by preventing MT-MT contact during locomotion and thus friction and the generation of torque within the bundle that would otherwise impart torsion to the neurite as a whole. It is tempting to speculate that these functions for tau are essential for mechanical neuroprotection throughout the nervous system.

How does loss of PTL-1 cause MTs to spiral around one another? The MT bundles structures we observe in *ptl-1* mutants are strikingly similar to bundles of twisted filaments predicted to assemble from chiral biological polymers (*Grason and Bruinsma, 2007*), suggesting similar physical mechanisms account for formation of twisted MTs bundles in neurons. We propose that the observed torque is an emergent property of MT assembly that appears in the absence of PTL-1, which normally mediates long-range repulsion between adjacent MTs. In the absence of such repulsion, MTs twist around one another so as to maximize the cohesive energy of a chiral bundle. The gain in binding energy per unit length comes in expense of an energy cost to bend individual MT around the central, longitudinal axis. This manifests as an external torque that twists the bundle. This argument applies both to bundles of chiral filaments like MTs and bundles of achiral filaments, since the binding energy gained as filaments come together is sufficient to deform individual filaments (*Cajamarca and Grason, 2014*). Axial compression of this twisted bundle, as occurs during locomotion, could potentially deform the whole axon (as seen in the elastic thread of *Video 3*) via cross-links to the plasma membrane. Although all microtubules possess a helical surface lattice (*Hunyadi et al., 2007*), this mechanism is expected to be independent of the number of protofilaments per microtubule, and to apply to both the unusual 15-protofilament MTs present in *C. elegans* TRNs, and also to the more common 13-protofilament MTs found in mammalian neurons. Consistent with this idea, neurons containing 13-protofilament MTs bend and deform into similar defects (as observed by us in TRNs) under compressive forces in cultured hippocampal neurons (*Roland et al., 2014*) and form twisted, hexagonal bundles in certain in vitro preparations (*Needleman et al., 2004*). In summary, we propose that MT bundles and actin-spectrin networks act synergistically to prevent shape defects, as a deficiency in both structures was required for the appearance of plectonemes. These structures and their molecular constituents ensure that axial tension, bending stiffness, and torque are balanced in manner sufficient to stabilize axons against mechanical deformations.

Spectrin, microtubules, and tau are among the most abundant components of the neuronal cytoskeletal and these proteins are conserved in all animals, from *C. elegans* to humans (*Baines, 2009*; *Findeisen et al., 2014*). The same mechanisms that protect *C. elegans* peripheral neurons against mechanical deformation may also protect axons in the mammalian central nervous system from damage caused by mechanical stress during daily life or during traumatic brain injuries. Intriguingly, mechanical stress has also been implicated in neurodegenerative disease (*Levy Nogueira et al., 2016*; *Wostyn et al., 2010*). Thus, it may not be surprising that pathological biomarkers such as loss of tau function (*Puvenna et al., 2016*) and spectrin degradation are shared between mechanical stress-based brain disease and aging-related neurodegeneration (*Cai et al., 2012*). Indeed, this work suggests the intriguing possibility that defects in the genes encoding tau, spectrin, and microtubule isoforms expressed in the nervous system sensitize axons to mechanical stress and function as risk factors in traumatic brain injury or neurodegenerative disease.

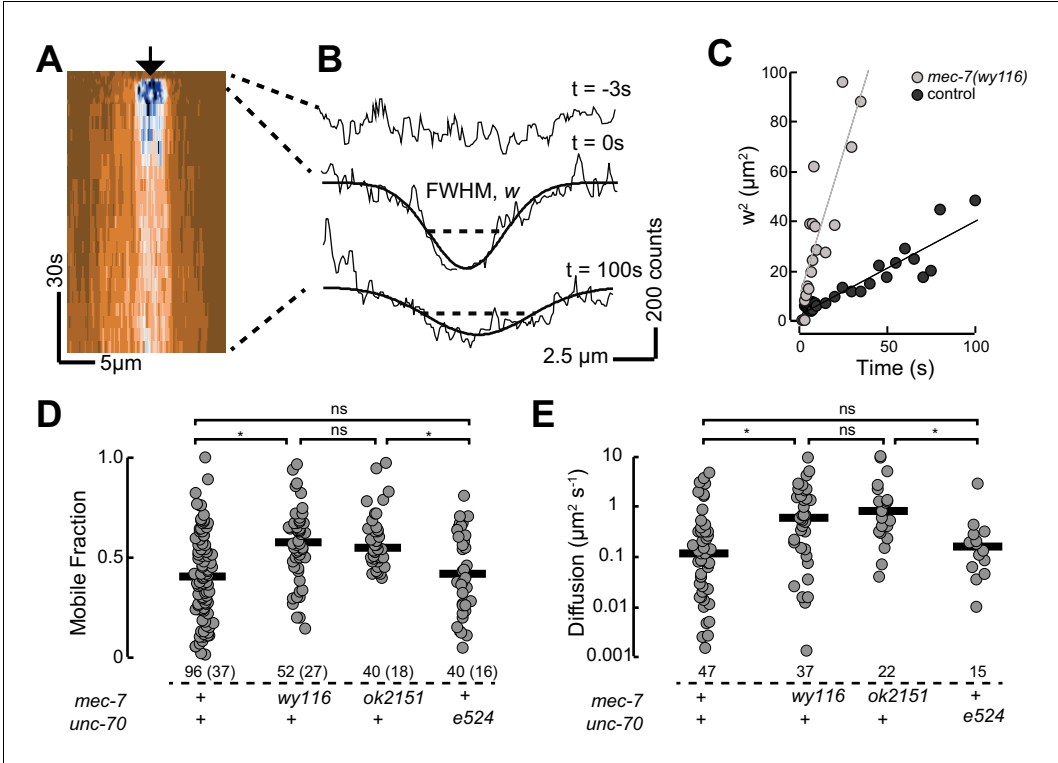

**Figure 8.** PTL-1 is more mobile in *mec-7(wy116)* mutants than control. (**A**) Representative kymograph of PTL-1:: mNG expressing control animals. The arrowhead points to the position, time, and bleaching area (2 μm wide). The color scale encodes low fluorescence in blue and high fluorescence in ochre. (**B**) Intensity profiles $t$ = 0.7, 2.8, and 21 s after bleaching. Smooth lines are Gaussian fits used to estimate the width of the bleached area. Diffusion of PTL-1::mNG into the bleached area results in an increase of the full-width at half maximum (FWHM). (**C**) Plot of squared FWHM vs. time. Smooth lines are fits to the data and their slope = *2D*, where *D* is the apparent diffusion constant, which had values of 1 μm$^2$ s$^{-1}$ ($n$ = 47) and 3 μm$^2$ s$^{-1}$ ($n$ = 37) for control and *mec-7(wy116)* animals, respectively. (**D**) Mobile fraction of PTL-1::mNG in the indicated genotypes. Numbers of FRAP curves and (animals) used for each experiment. Asterisks indicate statistically significant differences according to Dunn-Holland-Wolfe non-parametric test for multiple comparisons. (**E**) Diffusion constant of PTL-1::mNG in the indicated genotypes for a subset of the data from panel **D**). FRAP experiments were performed in both ALM and PLM neurons in vivo.

## Materials and methods

### *C. elegans* strains and touch assays

Strains used in this study were derived for this study, obtained from the Caenorhabditis Genetics Center or donated by colleagues (*Table 3*). Double mutants were obtained by genetic crosses, according to standard *C. elegans* procedures (*Stiernagle, 2006*). We used the *uIs31 [mec-17p::GFP]* transgene to visualize TRNs (*O'Hagan et al., 2005*); *oxIs95 [myo-2p::GFP; pdi-2p::UNC-70)]* to express wild-type UNC-70 β-spectrin in the epidermis (*Hammarlund et al., 2007*); *pgIs22 [unc-70p:: UNC-70(N-TSMod)]* to mark the N-terminus of UNC-70 (*Kelley et al., 2015a*); and *wyIs97 [unc-86p::myrGFP, unc-86p:: mCherry::RAB-3]* to visualize synaptic vesicles in the touch receptor neurons (*Richardson et al., 2014*). Transgenic animals expressing SPC-1::GFP fusion were obtained from the transgenome project (*Sarov et al., 2012*) (Gift of M. Sarov). Animals expressing PTL-1::mNeonGreen were generated de novo using CRISPR/Cas9 genome editing (*Dickinson et al., 2015*) to yield the *pg73* allele that expressed a fusion protein from the endogenous genetic locus. Touch sensitivity was assayed by a ten-trial method, as described (*Chalfie et al., 2014*). Animals were tested blind to genotype in at least two independent cohorts of 25 young adults, following synchronization by hypochlorite treatment (*Stiernagle, 2006*). All experiments in this study used young adult hermaphrodites.

**Table 2.** Microtubule numbers and length in wild type and mutant touch receptor neurons. Values are mean ± standard deviation for the number of neurons indicated in parentheses. Serial-section TEM data sets consisted of between 38 and 68 thin sections covering between 1.9 and 3.4 μm in length. Length is estimated from the number of microtubules per cross-section ($N$), series length (a) and the number of endpoints ($T$) detected in the series: $l = 2 \cdot N \cdot a / T$ (**Chalfie and Thomson, 1979**). A one-way ANOVA on the ranks (Kruskal-Wallis test) shows a significant effect ($\alpha = 0.05$) of genotype on microtubule number ($p=0.0465$) and length ($p=0.0420$) in ALM, but no effect of genotype on microtubule number ($p=0.827$) and length ($p=0.232$) in AVM. *$p<0.05$ different from control, Mann-Whitney test.

| Genotype | Neuron | Number | Length, μm | Source |
|---|---|---|---|---|
| N2 (Bristol) | AVM | 12.8 ± 3.3 (8) | 9.40 ± 0.4 (3) | *Chalfie and Thomson (1979)* |
| control, uIs31 [mec-17p::GFP] | AVM | 7 ± 3.6 (3) | 4.1 ± 0.6 (3) | This study |
| unc-70 (e524); uIs31 | AVM | 8 ± 2 (3) | 3.8 ± 0.4 (3) | This study |
| unc-70 (s1502); uIs31 | AVM | 23 ± 10 (5) | 2.7 ± 0.7 (2) | This study |
| N2 (Bristol) | ALM | 28.6 ± 4.1 (9) | 22.2 ± 3.6 (5) | *Chalfie and Thomson (1979)* |
| N2 (Bristol) | ALM | 57 ± 11 (3) | 14 ± 4.3 (3) | This study |
| control, uIs31 | ALM | 31 ± 2 (4) | 16 ± 4.5 (4) | This study |
| unc-70 (e524); uIs31 | ALM | 17.9 ± 3.4 (4)* | 5.3 ± 1.3 (4) | This study |
| unc-70 (s1502); uIs31 | ALM | 24 ± 11 (5) | 2.8 ± 0.9 (5) * | This study |
| mec-7 (wy116); wyIs97; uIs31 | ALM | 27 ± 13 (4) | 11 ± 6 (4) | This study |
| ptl-1 (ok621); uIs31 | ALM | 20 ± 10 (5) | 6.95 ± 6.6 (5) | This study |
| ptl-1 (ok621);unc-70(e524); uIs31 | ALM | 32 ± 0.8 (2) | 8.9 ± 3.5 (2) | This study |
| mec-7 (wy116);unc-70(e524); uIs31 | ALM | 44 ± 4 (3) | 8.9 ± 3 (3) | This study |

## Animal maintenance and transgenesis

Animals were maintained according to standard protocols (*Stiernagle, 2006*) and grown at 20°C, unless indicated otherwise. For this study, we analyzed young adult hermaphrodites who were age-synchronized using hypochlorite treatment (*Stiernagle, 2006*). To create *ptl-1(pg73*[PTL-1::mNeon-Green]), we used a self-excising cassette variant of CRISPR/Cas9 genome editing (*Dickinson et al., 2015*) to insert mNeonGreen (*Shaner et al., 2013*) into the genomic locus encoding PTL-1 after the last coding exon. The guide RNA sequence was: 5'-ACGTGTCGACGGACTCGAGC-3' and cloned

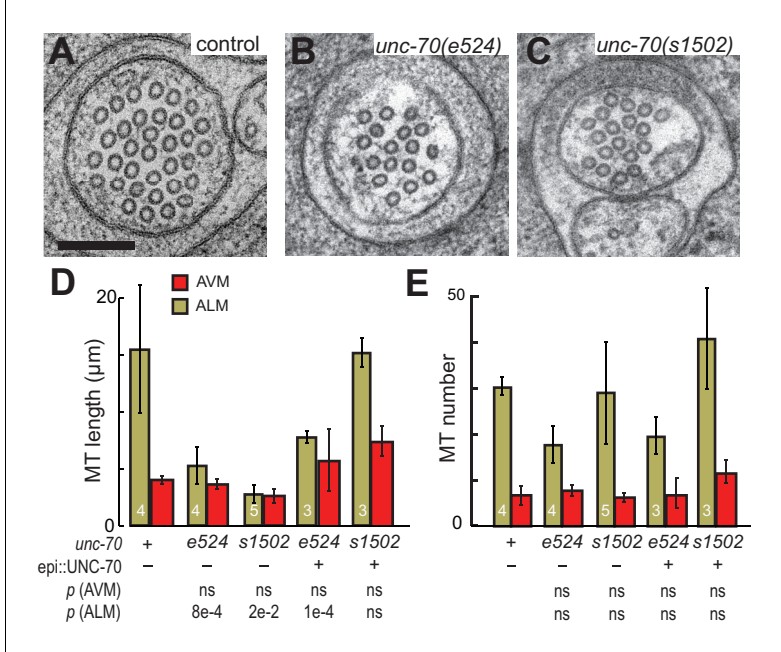

**Figure 9.** The spectrin cytoskeleton stabilizes microtubule bundles. (A–C) Representative transmission electron micrographs of control, TRN::GFP animals (A), *unc-70(e524)* missense mutants (B) and *unc-70(s1502)* null mutants (C). All animals carry the TRN::GFP transgene. (D) Microtubule length as a function of the indicated genotype in AVM and ALM neurons, including transgenic strains in which wild-type UNC-70 was expressed under the control of an epidermal-specific promoter, epi::UNC-70. A two-way ANOVA of these data revealed a significant effect of both cell-type [$F_{(1, 24)}$=16.54, p=0.0004] and genotype [$F_{(4, 24)}$=8.401, p=0.0002] and the results of post-hoc comparisons ($\alpha$ = 0.05) to the control condition are indicated below (ns = not significant with a *p*-value greater than 0.3). (E) Microtubule number as function of the indicated genotype in AVM and ALM neurons. TEM sections that overlapped with support grids were excluded from analysis. According to a two-way ANOVA, there was a significant effect of cell-type [$F_{(1,25)}$=28.94, p<0.0001] but not genotype [$F_{(4, 25)}$=2.139, p=0.1057] on microtubule number. Post-hoc comparisons ($\alpha$ = 0.05) to the control genotype found no significant differences (ns = not significant with a p-value greater than 0.3), as indicated.

The following figure supplements are available for figure 9:

**Figure supplement 1.** Microtubule number as a function of distance.

**Figure supplement 2.** Postbuckling stress distribution in AVM TRN.

---

into pDD162 (*Dickinson et al., 2013*) by site directed mutagenesis. Assembly of the repair template for homologous recombination was performed using Gibson assembly using PCR fragments directly amplified from genomic DNA with the following primers:

5' homology FWD:
acgttgtaaaacgacggccagtcgccggcaCGGAAATGTGGAAATTTTCTCG
 5' homology REV:
CATCGATGCTCCTGAGGCTCCCGATGCTCCTAACGAGCTGATGTCCAGCGTA
 3' homology FWD:
CGTGATTACAAGGATGACGATGACAAGAGATGATCGGACCACGCTTTCACT
 3' homology REV:
ggaaacagctatgaccatgttatcgatttcTAGAATAGCTCTCATCAAAGTAGTCCCAC

We injected 30 animals as described (*Dickinson et al., 2015*) and isolated 10 roller animals in the F2 generation, heat-shocked F3 animals from plates that showed an absence of wild-type animals, and verified insertion of PTL-1::mNeonGreen by PCR and by expression of mNeonGreen in a pattern similar to that reported previously (*Gordon et al., 2008*).

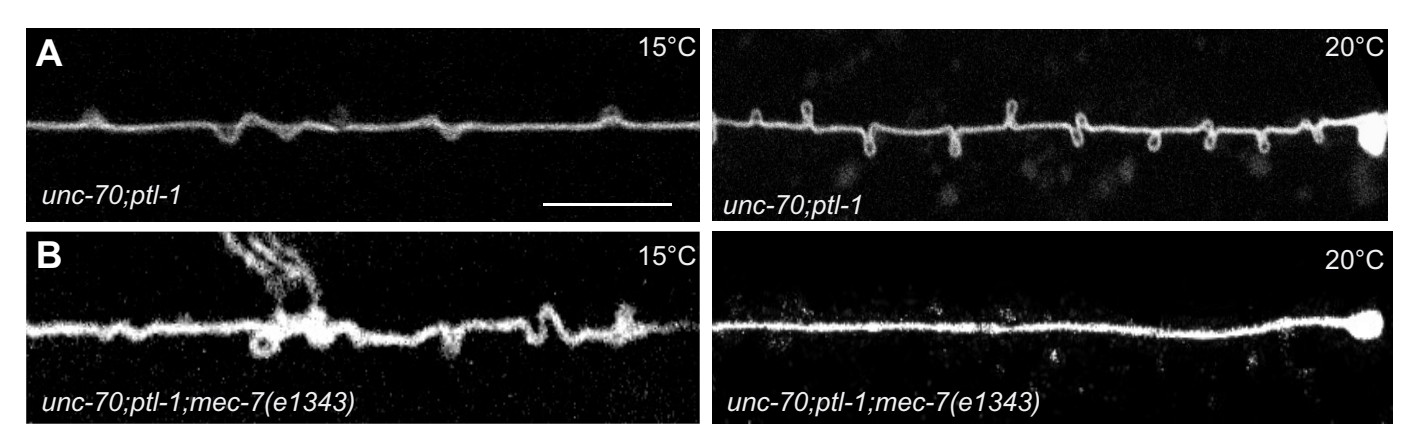

**Figure 10.** TRNs resemble a twisted thread in *ptl-1* mutants. (**A**) *unc-70(e524); ptl-1(ok621)* double mutant ALM TRNs grown at 15°C (left) and 20°C (right). Similar results observed in at least five animals per condition. Scale = 10 µm. (**B**) *unc-70(e524); ptl-1(ok621); mec-7(e1343ts)* triple mutant ALMs animal grown at 15°C (left) and 20°C (right). More loops are seen in the permissive temperature of 15°C than at the restrictive temperature of 20°C. Similar results observed in at least 10 animals/condition.

## Primary cell culture

Isolation of embryonic cells was performed as described (*Christensen et al., 2002*; *Krieg et al., 2014*; *Strange et al., 2007*). We plated cells on glass bottom dishes (thickness #1.5, WillCo wells), culture medium was exchanged daily, and cells were analyzed cells after 3 to 7 days of growth in vitro. We prepared cells for immunofluorescence using a 5-min treatment with Krebs fixative containing 4% PFA (Electron Microscopy Sciences; vol/vol) at room temperature (RT) followed by a 15-min treatment in ice-cold methanol at −20°C. Next, we treated cells in antibody buffer (1xPBS +0.01% Triton X100 +3%BSA weight/vol) for 1 hr at RT, exposed them to a primary antibody raised against GFP (rabbit-αGFP; Life Technology) for 6 hr at RT with gentle agitation, and washed them three times in PBST for a 15 min. The primary antibody was diluted 1:100 from a stock solution (1 mg/ml) stored at −20°C, according to manufacturer's instructions. We applied the secondary antibody (goat-anti-rabbit AbberiorStar 488, Abberior, 1:200 dilution) overnight at 4°C and washed for 4 hr at RT before mounting in Prolong Diamond antifade medium (LifeTechnology). Monoclonal antibody against acetylated tubulin was used at 1:2000 to identify TRNs in culture (SigmaAldrich, 6-11B, 6 hr at RT) and detected with Alexa-680 secondary antibody (goat-anti-mouse, Invitrogen, 1:2000, 4 hr at 4°C).

## AFM mechanics measurements

AFM experiments were carried out as described previously (*Krieg et al., 2014*; *Vasquez et al., 2014*). In brief, we mounted Biolever cantilevers (Olympus, 6 mN/m) on a biological AFM (Catalyst/Resolve, Bruker) for indentation tests and their spring constant was measured using the thermal noise method (*Hutter and Bechhoefer, 1993*). Cells were dissociated from TU2769 *uls31[mec-17p:: GFP]*, GN400 *uls31;unc-70(e524)*, GN119 *uls31 ptl-1(ok621)*, GN115 *uls31; mec-7(wy116)*, and GN643 *uls31 ptl-1(ok621);unc-70(e524)* animals. TRNs were identified in these mixed cultures by their expression of GFP. Force-indentation (*F-d*) curves were taken at 7 µm·s$^{-1}$ with a contact force of 500 pN. The contact geometry was modeled as a Hertzian impact between an incompressible sphere indenting the axon as a cylinder (*Puttock and Thwaite, 1969*). We fit *F-d* curves between 0 and 100 nm indentation (*Figure 1-Supplement 1)Figure 1—figure supplement 1*) according to $F = a \cdot d^{1.5}$, with *a* as the only free parameter. We used the fitted value of *a* to derive the apparent Young's modulus of the axon according to *Equation 7*.

$$E_{Axon} = -\frac{9 \cdot a \cdot E_b \cdot K1^{1.5}}{(9 \cdot a \cdot K1^{1.5}) - (\pi \cdot 8 \cdot E_b \cdot \sqrt{(d \cdot K2)})} \tag{7}$$

**Table 3.** Strains and transgenes used in this study.

| Strain | Genotype | Source or citation | RRID |
|---|---|---|---|
| N2 Bristol | N2 | CGC | WB_RWK40N2 |
| TU2769 | uIs31 [mec-17p::GFP] | (*O'Hagan et al., 2005*) | WB_TU2769 |
| CB524 | unc-70(e542) | CGC | WB_CB524 |
| EG4492 | unc-70(s1502); oxIs95 [myo-2p::GFP; pdi-2p::UNC-70] | (*Hammarlund et al., 2007*) | N/A |
| CB1343 | mec-7(e1343) | CGC | N/A |
| RB1708 | mec-7(ok2152) | CGC | WB_RB1708 |
| TV1838 | wyIs97 [unc-86p::myrGFP, unc-86p::mCherry::RAB-3] | (*Richardson et al., 2014*) | N/A |
| TH230 | WRM0620A_C08(pRedFlp-Hgr)(spc-1[28275]::S0001_pR6K_Amp_2xTY1ce_EGFP_FRT_rpsl_neo_FRT_3x-Flag)dFRT::unc-119-Nat | (*Sarov et al., 2012*) | WB_TH230 |
| GN061 | him-4(e1267); uIs31 | This study | N/A |
| GN400 | unc-70(e524); uIs31 | (*Krieg et al., 2014*) | N/A |
| GN401 | unc-70(s1502); uIs31; oxIs95 | (*Krieg et al., 2014*) | N/A |
| GN444 | unc-70(e524); uIs31; oxIs95 | (*Krieg et al., 2014*) | N/A |
| GN600 | pgIs22 [unc-70p:: UNC-70(N-TSMod)]; oxIs95 | (*Kelley et al., 2015*) | WB_GN600 |
| GN507 | him-4(e1267); unc-70(e542) | (*Krieg et al., 2014*) | N/A |
| GN514 | unc-70(s1502); uIs31 | (*Krieg et al., 2014*) | N/A |
| GN643 | ptl-1(ok621)uIs31;unc-70(e524) | This study | N/A |
| GN644 | uIs31; unc-70(e524); mec-7(wy116) | This study | N/A |
| GN119 | ptl-1(ok621)uIs31 | This study | N/A |
| GN115 | uIs31; mec-7(wy116) | This study | N/A |
| GN645 | ptl-1(ok621)uIs31; mec-7(e1343ts) | This study | N/A |
| GN704 | WRM0620A_C08(pRedFlp-Hgr)(SPC-1[28275]::S0001_pR6K_Amp_2xTY1ce_EGFP_FRT_rpsl_neo_FRT_3x-Flag)dFRT::unc-119-Nat; unc-70(e524) V | This study | N/A |
| GN658 | WRM0620A_C08(pRedFlp-Hgr)(SPC-1[28275]::S0001_pR6K_Amp_2xTY1ce_EGFP_FRT_rpsl_neo_FRT_3x-Flag)dFRT::unc-119-Nat; unc-70(s1502) V; oxIs95 IV | This study | N/A |
| GN659 | WRM0620A_C08(pRedFlp-Hgr)(SPC-1[28275]::S0001_pR6K_Amp_2xTY1ce_EGFP_FRT_rpsl_neo_FRT_3x-Flag)dFRT::unc-119-Nat; uIs31 | This study | N/A |
| GN660 | WRM0620A_C08(pRedFlp-Hgr)(SPC-1[28275]::S0001_pR6K_Amp_2xTY1ce_EGFP_FRT_rpsl_neo_FRT_3x-Flag)dFRT::unc-119-Nat; ptl-1(ok621) III | This study | N/A |
| GN655 | ptl-1(pg73[PTL-1::mNeonGreen]) | This study | N/A |
| GN647 | ptl-1(pg73); mec-7(ok2152) | This study | N/A |
| GN648 | ptl-1(pg73); unc-70(e524) | This study | N/A |
| GN649 | ptl-1(pg73); mec-7(wy116) | This study | N/A |
| GN705 | ptl-1(ok621) uIs31; unc-70(e524); oxIs95 | This study | N/A |

Using the following values: $E_b$ = Young's modulus of the cantilever (Si = $3.5 \cdot 10^9$ Pa); $K1$ = complete elliptic integral of the first kind ~1.71; $K2$ = complete elliptic integral of the second kind ~0.89; $d$ = diameter of the tip ~$60 \cdot 10^{-9}$ nm; $K1$ and $K2$ are from *Case 11: Sphere in contact with a cylinder (external)* in **Puttock and Thwaite (1969)**. The tip diameter was modeled as a round sphere using the nominal specifications given by the manufacturer: $D_s$ = 60 nm.

## Super-resolution STED imaging

We used a commercial STED instrument mounted on a Leica SP8 confocal microscope to analyze the subcellular organization of native actin-spectrin networks. We excited fluorophores on anti-GFP

antibodies with a pulsed 80 MHz white light laser at 488 nm and depleted them with a continuous 592 nm laser through a 100x/1.4 oil immersion objective lens. The depletion laser power was adjusted between 30 and 40% of its maximum output power. A time gate was used to reject photons with a lifetime outside a 1.3 and 6 ns time window. To achieve optimal sampling at the Nyquist frequency of the expected resolution (~50–70 nm), the pixel size was matched to be ~20 nm. The scan rate was set to 100 Hz, while averaging each line 8–16 times. All STED imagings were performed on fixed samples of cultured neurons (**Figure 5**, **Figure 5—figure supplement 2**).

## Confocal microscopy

Animals carrying the *uls31[mec-17p::GFP]* marker to visualize TRNs were prepared for confocal microscopy as described (**Krieg et al., 2014**) using a low percentage agar embedding technique (**Kim et al., 2013**). Animals carrying the *him-4* mutation have defects in cell-cell and cell-matrix junctions and some mutants have severe defects in body morphology (**Vogel and Hedgecock, 2001**). For this study, we selected *him-4* mutants whose gross morphology was essentially wild type. Under these conditions, animals undergo dorso-ventral body bends without moving out of the field of view. Animals were imaged on a Leica SP5 confocal microscope through a 20x/1.0 water immersion lens at a frame rate of 0.66 Hz. Fluorescence was excited using a 488 nm laser, and emission was collected between 500 and 600 nm. The recording was stopped after ~100 frames. We used these videos to measure the normalized length and curvature of the ALM and AVM process as function of body posture (**Figure 2**, **Figure 2—figure supplement 1**).

## Axon shape analysis

We analyzed neuron shape in living animals as follows. First, we imaged young adult animals carrying the *uls31*[*mec-17p*::GFP] transgene and selected segments for further processing. Next, we corrected for body curvature using polynomial spline fitting in Fiji (**Schindelin et al., 2012**), binarized the images and exported them into IgorPro (Wavemetrics, Oswego, OR) after finding the centerline of the neuron using the standard skeletonize procedure. Finally, we analyzed its path by computing each neurite's deviation $D$ from the straightest possible path via a directed walk along the long (anterior-posterior) axis of the neurite with a second random component along the dorsal-ventral axis. Through this approach, we derived a measure of the degree of deviation from a straight path from the mean-squared displacement (MSD) in the dorsal-ventral axis, $\langle x^2 \rangle$, over the length of axon according to **Equation 8**:

$$\langle x^2 \rangle = \frac{1}{N-(n+1)}\sum_{i=0}^{N-n-1}\left[(x_{n+i}-x_i)^2+(y_{n+i}-y_i)^2\right] \tag{8}$$

Here, $x$ is the displacement along the anterior-posterior or long axis and $y$ is the displacement along the orthogonal dorsal-ventral axis, $N$ is the number of samples (pixels or contour length encompasing the neuron), and $n$ is the rolling window size (e.g. in pixels or segmentlength). For each segment length $n$, we sum over all possible windows that fit into the contour of the neuron and normalize by the number of 'measurements' (or windows/segments). For a straight line, the error will be zero and the MSD will be the square of the contour.

Next, we plotted MSD versus $n$ and fit the curve to expected relationship:

$$\langle x^2 \rangle = S^2 n^2 + Dn \tag{9}$$

In **Equation 9**, $S$ is the persistence and $D$ is used as a measure of the deviation of the neuron from a perfectly straight path (*i.e.* As $D$ approaches zero, the relationship between MSD and $n$ approaches that expected for a parabola.)

## Fluorescence microscopy after photobleaching (FRAP)

*Experimental:* We used a confocal microscope (Leica SP8) to perform FRAP experiments on *ptl-1 (pg73*[PTL-1::mNeonGreen]) III. A small region-of-interest was bleached by scanning the area at 100% power with the 488 nm line of an Argon ion laser. Prior to and following the bleaching phase, power was attenuated to 0.5–2%. Three images were acquired at 1.428 Hz prior to bleaching and the recovery from photobleaching was monitored at 1.428 Hz for 10 frames and 0.2 Hz thereafter. This strategy enabled us to capture the initial and rapid recovery from bleaching.

*Theory:* The recovery process was modeled as diffusive process in one dimension, with an infinite reservoir of emitters (fluorophores) inside the axon. To analyze the recovery process, we follow the approach outlined by *Hauser et al. (2008)*, applying a one-dimensional diffusion equation: $\frac{\partial C}{\partial t} = D\frac{\partial^2 C}{\partial x^2}$ with $C$ as the concentration of the emitters (fluorophores), $t$ as the time and $x$ as the direction of the diffusing species (corresponding to the orientation of the axon) and $D$ as the diffusion constant. The solution has a Gaussian shape: $C(x,t) = \frac{M}{2\sqrt{(\pi D t)}} e^{-x^2/4Dt}$, with $M$ the total amount of the diffusing species, which is, in the case of an FRAP experiment, the bleached emitters diffusing out of the bleach spot. It is readily apparent that the dispersion of the Gaussian profile, $w(t)$, is related to the diffusion constant according to $w^2 = 2Dt$. Thus, it is possible to determine the diffusion coefficient by plotting $w^2$ as a function of time for a series of intensity profiles obtained from images taken during the recovery process and straight line with slope $2D$ passing through the origin is expected. Similarly, the change in amplitude of the Gaussian bleach profile over time is related to the dimensionality of the recovery process (*Hauser et al., 2008*).

*Data processing* was performed in IgorPro (Wavemetrics) as follows:

1. A previously described algorithm (*Krieg et al., 2014*) was applied to automatically trace the axon in the image and collect the intensity *I* values along its contour *x*.
2. The *I(x)* data was fit to a 1D Gaussian by the least-squares method to extract the width, $w^2$, and amplitude as a function of time for each image.
3. To estimate *D* from the recovery process, the fitting parameter $w^2$ is plotted vs. *t*, which yields a straight line of slope *2D*.
4. To estimate the mobile and immobile fraction, the intensity values of the bleached region was normalized according to $I(t) = \frac{I_{t=\infty} - I_{t=0}}{I_{t<b} - I_{t=0}}$ with $I_{t=0}$ as the intensity directly after bleaching, $I_{t=\infty}$ as the intensity after recovery and $I_{t<b}$ as the pre-bleach intensity. *I(t)* was plotted against *t* and fit to a single exponential using $I(t) = I_{t=0} + I_{t=\infty} e^{(t/\tau)}$.

## Isolation of the *mec-7(wy116)* allele

The *mec-7(wy116)* allele was isolated in genetic screen for mutants displaying defects in synaptic vesicle transport in the touch receptor neurons. We used TV1838 *wyIs97 [unc-86p::myrGFP, unc-86p::mCherry::RAB-3]* transgenic animals (*Richardson et al., 2014*) to visualize synapses in the TRNs. *wyIs97* drives expression of membrane-bound GFP (myrGFP) and the vesicle protein RAB-3 tagged by mCherry and reveals an accumulation of synaptic vesicles in a distal branch of the PLM neuron in the ventral nerve cord. Briefly, *wyIs97* animals were mutagenized by exposure to ethane methanesulfonate (EMS) and screened for a loss of mCherry-tagged synaptic vesicles in the ventral cord synapses formed by the PLM neurons. The intensity of the synaptic patch was reduced nearly 50-fold in *wy116* mutants compared to wild-type controls. The *wy116* allele was mapped to the genomic interval containing *mec-7* using standard genetic mapping techniques and sequenced to identify a missense, molecular defect replacing the threonine at position 409 with an isoleucine.

## Serial-section transmission electron microscopy (ssTEM)

We prepared wild-type, control, and mutant animals prepared for transmission electron microscopy (TEM) using high-pressure freezing (HPF) followed by freeze substitution (FS), as previously described (*Cueva et al., 2012*, *2007*). The following strains and genotypes were used for ssTEM analysis: wild-type, N2 (Bristol); control, TU2769 *uIs31[mec-17p::GFP]* III; mutant, GN115 *uIs31; mec-7(wy116)*; GN400 *uIs31; unc-70(e524)*; GN514 *uIs31; unc-70(s1502)*; GN119 *uIs31ptl-1(ok621)*; GN643 *uIs31ptl-1(ok621);unc-70(e524)*; GN644 *uIs31; unc-70(e524); mec-7(wy116)*; EG4492 *unc-70 (s1502);oxIs95*. Briefly, we used an AFS freeze substitution (Leica, Germany) apparatus to preserve worms in either 0.2% glutaraldehyde (GA) and LR-White or 1% $OsO_4$ plus 1% Uranyl acetate in acetone with or without 3% water (*Walther and Ziegler, 2002*) and embedded in Epon. Serial, ultrathin sections (50 nm) were cut with a Diatome diamond knife on Leica Ultracut S or T microtomes and collected on Formvar-coated, slot grids. To enhance contrast, sections were poststained in 3.5% uranyl acetate in acetone for 30 s, and then Reynold's lead citrate preparation for 3 min, or 5% aqueous uranyl acetate for 15 min followed by Sato's lead for 3 min. Grids were examined on JEOL (Tokyo, Japan) TEM 1230 and 1400 transmission electron microscopes and imaged with Gatan 967 or 833 CCD cameras (Gatan, Pleasonton, CA).

## Analysis of serial-section EM data sets

We collected 60–120 consecutive thin-sections covering 3–5 μm, imaged these sections and used them to construct three-dimensional (3D) models using Reconstruct (*Fiala, 2005*) or TrakEM2 and IMOD (*Cardona et al., 2012*; *Kremer et al., 1996*). Briefly, images from consecutive sections were aligned manually and the outline of all subcellular structures traced by hand. The average microtubule length was calculated from $L = 2 N a/T$, where $N$, $a$, and $T$ are the average number of microtubules per section, the length of the series, and the number of microtubule terminations in the series (*Chalfie and Thomson, 1979*). Microtubule bundle organization (angles, inter-MT spacing) was quantified in ImageJ blind to genotype for at least seven sections and four neurons for each genotype. Oblique MT cross-sections were omitted from this analysis. TEM sections that overlapped with support grids were excluded from analysis.

The helical pitch of mutant MT bundles was analyzed by applying particle imaging velocimetry (PIV) to TEM image stacks. Images were preprocessed as follows and pairs of adjacent sections were analyzed using PIV plugin for Fiji (*Martiel et al., 2015*), relying on microtubule profiles as fiducial markers. The first step is to align serial section TEM images to generate an image stack. Next, the image stack was cropped to show only the TRN and its MT bundle and pairs of images were selected to represent the entire stack, generating a set of mini-stacks. Lastly, image contrast was inverted prior to PIV analysis such that microtubule profiles were the brightest points in the image, representing the particles to be analyzed. The PIV algorithm was applied to all mini-stacks with an interrogation window of 32 pixels$^2$ within a search window of 96 pixels$^2$. Microtubule displacement was determined by searching for the maximal intensity correlation between the interrogation and search windows, with a threshold of 0.6. [Lower values for the correlation threshold result in poor predictions in displacement vectors (*Martiel et al., 2015*).] The magnitude of the displacement vector was quantified as a function of the distance to the center of the bundle using the Radial Profile Extended plugin of FIJI and plotted in *Figure 7D*. The magnitude of the vector was then converted into an angle using the relation $\theta(r) = \tan^{-1}\left(\frac{r}{z}\right)$, in which theta is the helical pitch angle, $r$ is the displacement vector and $z$ is the section thickness of the two neighboring electron microscopy slices, in our case 100 nm.

## Mechanical modeling

We used a discrete elastic rod model (*Bergou et al., 2010*, *2008*, *2008*; *Jawed et al., 2014*; *Jawed and Reis, 2014*) to simulate neuron shape defects, including helices and plectonemes. This model derives discrete counterparts of the smooth equations of motion directly in the discretized domain and uses the concept of parallel transport, such that the reference frame stays adapted to the centerline, as presented previously (*Bergou et al., 2008*, *2010*; *Bobenko et al., 2008*; *Grinspun et al., 2006*). In our simulations, we modeled the neuron as an extensible rod subject to bending and twisting deformations. We assumed that axon diameter (ca. 200 nm) is small in comparison to the curvature radius and that the cross section of the TRN is isotropic. The model excludes external force fields such as gravity and assumes that viscous forces dominate over inertial ones. TRN axons were assumed to be straight and have no initial curvature. The dynamics of the model neuron can thus be deduced from the proper application of a force balance, yielding the equations of motion for linear and angular accelerations (*Bergou et al., 2010*).

We applied Euler-Kirchoff theory (*Goriely and Tabor, 1998*; *Nizette and Goriely, 1999*) to capture the physical stability of neuronal processes across a range of mechanical properties (*Bergou et al., 2010*, *2008*; *Jawed et al., 2014*; *Jawed and Reis, 2014*). In our simulation, the elastic energy of the axon is decomposed into bend, stretch and twist energies. These are described by a bending stiffness $B$ (*Equation 1*), an axial stiffness $k$ (*Equation 2*), and a torsional stiffness $C$ (*Equation 3*), which is modeled with a shear modulus $G$ (*Equation 4*) with the Young's modulus $(E)$ and Poisson's ratio $\nu$.

The Kirchhoff rod is represented by an adapted framed curve, which is tangent-aligned (***t***) with the rod's centerline. To model a twist deformation, an orthonormal material frame defined by the vectors ***d₁*** and ***d₂*** is assigned to each point on the centerline such that $d_1 \times d_2 = t$. ***d₁*** and ***d₂*** span the plane normal to the centerline's tangent, called the cross-section, while ***t*** lies tangent to the centerline. This material frame is adapted to the centerline and can rotate around the tangent and thus

can describe a twisted configuration. The elastic energy per unit length (strain energy density $H$) of a deformed configuration is described by the components bending, twist and stretch energy:

$$H = \mathcal{E}_{bend} + \mathcal{E}_{twist} + \mathcal{E}_{tension} \tag{10}$$

$$H = \left[\frac{1}{2}B(\kappa_1)^2 + \frac{1}{2}B(\kappa_2)^2\right] + \frac{1}{2}C(\bar{\tau}_0 - \tau_1)^2 + k_0(|\bar{l}_0 - l_1|) \tag{11}$$

where $\mathcal{E}_{bend}$, $\mathcal{E}_{twist}$, and $\mathcal{E}_{tension}$ are the energies due to bending, twisting, and stretching, respectively, $\kappa_1$ and $\kappa_2$ are the principle curvatures along the two principle axes spanning a plane perpendicular to the unit tangent vector (longitudinal axis) of the neuron, which are defined as $\kappa(s) = \frac{\partial^2 R}{\partial s^2}$. $R(s,t)$ is the dynamic space curve coordinate with $s$ as the material coordinate and $t$ = time. The tangent vector is defined as $t = \frac{\partial R}{\partial s}$. The natural torsion of the undeformed neurons is $\bar{\tau}_0$ and $\bar{l}_0$ is the undeformed length of the rod. Due to the stretch energy, an application of an end load (tension) has a stablizing effect.

We simulated friction with the extracellular environment as viscous drag using a Stoke's drag on a cylinder in a low Reynolds number regime for every line segment of the discretized rod. The simulations were implemented in C$^{++}$ and based on the implementation of *Bergou et al. (2010)* and *Jawed et al. (2014)* available at http://www.cs.columbia. edu/cg/elastic_coiling/. To evaluate how axon shape depended on axial stiffness, bending stiffness and shear modulus, we varied each modulus by a factor of ten. The range of values examined in the simulations were: (1) Bending stiffness: $1.25 \cdot 10^{-24}$ N m$^2$ < $B$ < $1.5 \cdot 10^{-23}$ N m$^2$ and (2) Young's modulus: 1 kPa < $E$ < 10 kPa, which defines values for $G$ and $k$ according to *Equations 2 and 4*, respectively. Using the approximation describing the tensile bulk modulus for classical rods $k = \frac{EA}{L}$, we found that these values map well in between the values for the tension measured previously using atomic force microscopy ( $\sim 2 \cdot 10^{-6}$ N/m).

## Statistical methods and data handling

For normally distributed datasets, the sample size needed to detect a difference was calculated given the variance in the experimental data. Otherwise, sample sizes were not predetermined. Experimental parameters were varied randomly, and no sequence in data acquisition was applied. Behavioral experiments were conducted blind to genotype.

For TEM, sample size was limited by the extensive labor required to for sample preparation, imaging serial sections, aligning the image stacks, and analyzing the distribution of microtubules. Microtubule organization (inter-MT distance, angular distribution) was analyzed blind to genotype. Variance of microtubule number in different genotypes was tested using Fisher's $F$-test. We used Kolmogorov-Smirnoff or Jarque-Bera methods to test if our data were normally distributed and the modified Levene's test of the median to test for equality of variances in non-normally distributed data.

## Acknowledgements

We thank Z Liao for expert technical support, including (but not limited to) many worm injections; J Mulholland and C Espenel of the Cell Sciences Imaging Facility (CSIF) for technical assistance with STED and AFM measurements; D Kamin, E D'Este and S Hell for help with STED imaging and enabling us to collect preliminary data in their laboratory; D Dickinson for plasmids encoding mNeonGreen. Some strains provided by M. Sarov or the Caenorhabiditis Genetics Center (CGC), which is funded by NIH Office of Research Infrastructure Programs (P40 OD010440). We thank S. Wegmann and A. Goriely for critical comments on the manuscript; named and anonymous reviewers for their attention, feedback and detailed recommendations; and members of the Goodman, Dunn and Pruitt laboratories for fruitful discussions. Work funded by grants from NIH: R01NS092099 (MBG) and K99NS089942 (MK); ERC Consolidator Grant '3D Reloaded (JS and DC); HHMI Investigator Award (KS); and a HHMI Faculty Scholar Award (ARD).

## Additional information

### Competing interests

KSh: Reviewing editor, *eLife*. The other authors declare that no competing interests exist.

### Funding

| Funder | Grant reference number | Author |
| --- | --- | --- |
| National Institute of Neurological Disorders and Stroke | R01NS092099-02 | Alexander R Dunn<br>Miriam B Goodman |
| National Institute of Neurological Disorders and Stroke | 5K99NS089942-02 | Michael Krieg |
| Howard Hughes Medical Institute | | Kang Shen<br>Alexander R Dunn |
| H2020 European Research Council | ERC-2014-CoG | Jan Stühmer<br>Daniel Cremers |

The funders had no role in study design, data collection and interpretation, or the decision to submit the work for publication.

### Author contributions

MK, Conceptualization, Formal analysis, Supervision, Funding acquisition, Validation, Investigation, Visualization, Methodology, Writing—original draft, Writing—review and editing; JS, Software, Formal analysis, Investigation, Visualization, Methodology, Writing—original draft, Writing—review and editing; JGC, Formal analysis, Validation, Investigation, Visualization, Methodology; RF, Data curation, Formal analysis, Validation, Investigation, Visualization, Methodology, Writing—original draft, Writing—review and editing; KSp, Resources, Investigation; DC, Resources, Software, Supervision, Funding acquisition, Writing—review and editing; KSh, Resources, Funding acquisition, Investigation, Writing—review and editing; ARD, Conceptualization, Supervision, Funding acquisition, Validation, Visualization, Project administration, Writing—review and editing; MBG, Conceptualization, Resources, Data curation, Formal analysis, Supervision, Funding acquisition, Validation, Visualization, Methodology, Writing—original draft, Project administration, Writing—review and editing

### Author ORCIDs

Michael Krieg, http://orcid.org/0000-0003-0501-5036

Kang Shen, http://orcid.org/0000-0003-4059-8249

Alexander R Dunn, http://orcid.org/0000-0001-6096-4600

Miriam B Goodman, http://orcid.org/0000-0002-5810-1272

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
