## [Decision Letter]

Thank you for submitting your article "The Mechanics of Supercoiling in *Caenorhabditis elegans* Neurons" for consideration by *eLife*. Your article has been reviewed by two peer reviewers, and the evaluation has been overseen by Anna Akhmanova as the Senior and Reviewing Editor. The following individual involved in review of your submission has agreed to reveal his identity: Michel Labouesse (Reviewer #1).

The reviewers have discussed the reviews with one another and the Reviewing Editor has drafted this decision to help you prepare a revised submission.

Summary:

This manuscript combines quantitative modeling and high resolution imaging techniques (atomic force microscopy, STED, high-pressure serial electron microscopy imaging in the background of genetic mutants) to analyze how axons resist mechanical stresses. The authors reveal that axon shape is stabilized by combining the effects of three main actors: the actin-spectrin cytoskeleton protects against axial tension, microtubules against bending and the tau protein against torque. Their results bring significant novelty in understanding how mechanical constrains can affect axon shape and can be counteracted. Overall, the results are of extremely high quality. Conceptually, they bring a significant and important contribution to the understanding of how mechanical forces impact on the function of the nervous system.

Essential revisions:

This manuscript combines quantitative modeling and high resolution imaging techniques (atomic force microscopy, STED, high-pressure serial electron microscopy imaging in the background of genetic mutants) to analyze how axons resist mechanical stresses. The authors reveal that axon shape is stabilized by combining the effects of three main actors: the actin-spectrin cytoskeleton protects against axial tension, microtubules against bending and the tau protein against torque. Their results bring significant novelty in understanding how mechanical constrains can affect axon shape and can be counteracted. Overall, the results are of extremely high quality. Conceptually, they bring a significant and important contribution to the understanding of how mechanical forces impact on the function of the nervous system.

1) Title: The title doesn't really convey the key message. Please change the title for something more biologically appealing.

2) The model. First, the Results start in a rather abrupt way with the presentation of the model without much context. It would be important for the lay neurobiologist to give some background, as very few readers will be familiar with the work of Alain Goriely. Ideally, the authors should start with some biological observation, and from there go on with their model. Second, it would be much easier to have the final section of the model assumptions and discussion in the main text (section 4 of the Supplementary notes). In this section, it would be important to give the range of the shear modulus values (as done for tension and stiffness). Please correct the many typos in that section.

3) Figure 1: The legend reads: "leading to plectonemes spaced at regular intervals." This is not really demonstrated in the figure. This would require adding a zoomed-out example that would demonstrate regularly spaced plectonemes, as well as a graph that depicts spacing as a function of mechanical properties. This is somehow present in the color-coded data points, but these are too hard to read and do not demonstrate regular spacing. Further, while discussing the observation that plectonemes arise at regular intervals, the authors may want to compare their observations and model with the phenomenon of pearling instability, whereby changes in tension induce a chain of "pearls" at regular intervals w in an axon (Proc. Natl. Acad. Sci. USA Vol. 96, pp. 10140-10145, August 1999; PRL 96, 048104 2006).

4) Figure 2: The ruffling of the axon is very apparent in the cartoons, but less so in the actual images of the AVM neuron (e.g. AVM in Dii). Please provide zooms. Furthermore, since some of the live data is very striking, it would be good to show still images from the movies illustrating axonal shape in different orientations. The reversal of top-down orientation between e.g. Ai and Aii confusing (red axon at bottom or on top). Finally, the panels iii are not really discussed in the text.

5) Figure 3: These data suggest that defects in spectrin reduce microtubule length in the ALM, possibly due to fracture. However, it also shows that this effect can be rescued by expressing spectrin in the epidermis, i.e. non-cell autonomously. This suggests that axial stress is rescued in this condition. It would be very good to test that by STED imaging of *unc-70(e524)* in the presence of epidermal *unc-70*. In addition, do the axons still buckle when *unc-70* is expressed in the epidermis? Pictures/analysis as in Figure 2 should be shown for this situation.

Further, the authors show that epidermis expression of UNC-70 partially rescues the microtubule number and length in ALM neuron in the null mutation *unc-70(s1502)* but not in the tetramer domain mutant *unc-70(e524)*. Could the authors comment on that? An equally plausible explanation for that rescue could be that these animals are paralyzed, which as detailed in Hammarlund, Jorgensen and Bastianiet al., 2007) is sufficient to rescue some of the *unc-70* axon phenotypes.

Furthermore, why are MTs in the AVM not shorter upon *unc-70* mutation, even though these cells experience more mechanical deformations (Figure 2)? Does a length-dependent persistence length along published values really explain this? Strikingly, the average MT length in the AVM is also more than twice shorter than previously published. What is the explanation for this? Panel D of Figure 3 misses significance testing.

6) The periodicity of spectrin distribution is not very obvious in vivo. For instance, it is difficult to see α-spectrin periodicity in the ALM axon Figure 4—figure supplement 2, and the peak in the PSD graph is not obvious. Could the authors improve the data? Could the authors explain why there are two peaks in Figure 4—figure supplement 2? Besides, how do the authors explain that the localization of α spectrin is not perturbed in the *unc-70(e524)* mutant? Does expression of epidermal UNC-70 in the epidermis restore spectrin periodicity in axons?

7) Figure 6: The twisted microtubules in panel d are very striking. Since this is a very important aspect of the paper, it would be good if this observation can be quantitatively compared to the other conditions. What is the pitch of the twist? Could the authors explain in more detail how the angle between MTs was determined (a scheme would help), and how the twisting of the axon and MTs would arise? Does the observed twist depend on the animal moving, and would it be observed in a fully paralyzed mutant?

8) Figure 6/7: FRAP is discussed before a large fraction of Figure 6 is discussed. Please consider restructuring this part a bit. I.e. put 287-302 before the FRAP data.

9) Figure 8: what happens in *ptl-1* mutants that have epidermal expression of *unc-70*?

Further, laser cutting experiment is not at all convincing, since the torque release in the Video 8 and Figure 8 is difficult to see. Could the author provide clearer examples? Besides, the color-coded image is quite difficult to interpret. Please show separate stills from the movie and mention how often this was observed.

10) Data and Discussion on torque generation:

The authors present AFM data to show how stiffness is perturbed in *unc-70* mutants or after Latrunculin A treatment. The parameter space derived from their model predicts that the formation of plectonemes through torque depends on tension and stiffness. To substantiate why axons do not form plectonemes, despite the appearance of torque in *ptl-1(ok621)*, it would be important to determine the stiffness of *ptl-1* axons and/or *mec-7(wy116)* mutants (prediction would be that it should be normal). The Results section on *ptl-1* ends with a bold statement saying "organized into a twisted bundle, implying *ptl-1* mutant axons have both decreased bending stiffness and increased torque". What is the basis for this statement? It would help to have in the text the constitutive equation accounting for it (if based on an equation).

The origin of torque generation should be discussed better. The authors suggest that MTs twist to maximize the length of contact lines. This suggests that it is related to the microtubules having 15 protofilaments and therefore having a supertwist. Does the pitch of twisting correspond with the pitch of the protofilament supertwist? Does this mean that 13 pf do not show this effect and that these finding cannot be translated to mammalian axons?

11) Methods: It would be important to better explain in the main text how the shape analysis was quantified and done. The equation provided in the Methods is not very clear: (i) why sum from i=0 to i=N-n? (ii) my understanding of path length is that it measures a distance, and should thus be expressed in meters; if so, the <∆χ2> would overall be expressed in meters, correct?

---

## [Author Response]

*Essential revisions: […] 1) Title: The title doesn't really convey the key message. Please change the title for something more biologically appealing.*

We revised the title to encompass both the physical model for sensitizing axons to mechanical stress and the proteins we identify as key players. The new title is:

“Genetic Defects in β-spectrin and tau Increase Torque-Tension Coupling and Sensitize *C. elegans* Axonsto Movement-induced Damage”

*2) The model. First, the Results start in a rather abrupt way with the presentation of the model without much context. It would be important for the lay neurobiologist to give some background, as very few readers will be familiar with the work of Alain Goriely. Ideally, the authors should start with some biological observation, and from there go on with their model. Second, it would be much easier to have the final section of the model assumptions and discussion in the main text (section 4 of the Supplementary notes). In this section, it would be important to give the range of the shear modulus values (as done for tension and stiffness). Please correct the many typos in that section.*

As we understand it, this critique has two main components. First, the reviewers argue that the mechanical simulations (original Figure 1) are a difficult entry point for many readers and suggest that the presentation of results be reordered to begin with a biological observation. This is a good point for the readership of *eLife* and pushed us to re-order the empirical and computational work. As the reviewers and editors can imagine, responding to this request has resulted in significant revisions to the entire text and all of the figures.

The revised flow of ideas in the manuscript is as follows. The revised manuscript leads with the observed shape defects found in mutants carrying defects in actin-spectrin networks, the microtubule bundle, and both structures. We now provide both static and dynamic images of the most striking defects (Figure 1 and its supplements). Next, we proceed to discuss how axon position influences movement-induced mechanical stresses (Figure 2 and their supplements). The simulations reproducing coils and plectonemes are presented next (Figure 4 and its supplements). The remainder of the manuscript is devoted to our exploration of the genetic and structural basis of axial tension, bending stiffness, and torque (Figure 5–Figure 9). The manuscript closes with a test of the overall model showing that an intact microtubule is required for the formation of plectonemes. As in the initial submission, the work presented here relies on genetic dissection in a simple invertebrate and combines visualization and quantification of axonal shape defects, simulations of how axon-like structures respond and resist compression-induced local shape defects, investigations of the actin-spectrin networks by super-resolution microscopy and of microtubules by serial-section transmission electron microscopy.

Second, the reviewers asked for model assumptions and parameter values evaluated in the simulations (Section 4, Old ‘Supplementary Notes’) to be incorporated into the text. This is a fine way to help readers place the model, its assumptions, and values for physical parameters into context. We added the relevant information to the text (Results) and refrain from publishing the full presentations of the simulation strategy, which is explained in greater detail in its original publication by Bergou et al., 2008, 2010. The new text:

“In the classical description of rods, the mechanical parameters *B, C,* and *k* all depend on Young’s modulus, *E*, and the rod’s cross-sectional area, *A*, such that changes in one mechanical parameter also affect the others. […] Similar arguments can be made for shear modulus and tension, justifying the large parameter space explored in our simulations (Figure 4).”

*3) Figure 1: The legend reads: "leading to plectonemes spaced at regular intervals." This is not really demonstrated in the figure. This would require adding a zoomed-out example that would demonstrate regularly spaced plectonemes, as well as a graph that depicts spacing as a function of mechanical properties. This is somehow present in the color-coded data points, but these are too hard to read and do not demonstrate regular spacing.*

We agree with the spirit of this comment – namely that the previous presentation did not adequately show or quantify the spacing observed between plectonemes in real or simulated axons. In response, we added the following figures and analysis to the revised manuscript. First, we provide images of mutant axons showing examples of regular spacing in between plectonemes (Figure 1). Second, we measured the inter-plectoneme interval in *ptl-1;unc-70* mutant axons and report the results in the text: “The defects in *ptl-1;unc-70* mutants appeared at roughly regular intervals of 13 ± 0.5 µm (mean ± s.e.m., n = 121 intervals in 10 neurons) along the axon (Figure 1) and were dynamic.” Third,we compare inter-plectoneme intervals measured in real and simulated axons by showing representative images (Figure 4) and plotting the real and simulated inter-plectoneme intervals as a bar graph (Figure 4—figure supplement 3).

*Further, while discussing the observation that plectonemes arise at regular intervals, the authors may want to compare their observations and model with the phenomenon of pearling instability, whereby changes in tension induce a chain of "pearls" at regular intervals w in an axon (Proc. Natl. Acad. Sci. USA Vol. 96, pp. 10140-10145, August 1999; PRL 96, 048104 2006).*

We added a brief discussion of pearling instability to the Discussion:

“Changes in tension have also been linked to pearling, a phenomenon in which tubes under tension form structures that resemble pearls on a string (Bar-Ziv et al., 1999; Pullarkat et al., 2006). Whereas pearling is driven by a competition between membrane tension and actin rigidity (Bar-Ziv et al., 1999), the periodic defects we found in tau-spectrin double mutants occur due to an unbalanced torque-bend coupling, such that internal forces of the MTs are the driving force for the observed deformations.”

*4) Figure 2: The ruffling of the axon is very apparent in the cartoons, but less so in the actual images of the AVM neuron (e.g. AVM in Dii). Please provide zooms. Furthermore, since some of the live data is very striking, it would be good to show still images from the movies illustrating axonal shape in different orientations. The reversal of top-down orientation between e.g. Ai and Aii confusing (red axon at bottom or on top). Finally, the panels iii are not really discussed in the text.*

We thank the reviewers for their attention to our presentation of axonal shape defects. We revised the figure to conform to this very helpful comment, including adding zooms. At the same time, it also made the previous composition of the figure unwieldy. To accommodate these changes, we reorganized the information into two separate figures. Figure 2 (and its supplements) show AVM and ALM axon curvature in control and *him-4* mutants, illustrating the impact of the axon position on mechanical stress. (The *him-4* mutation disrupts hemidesmosomes that attach the ALM neurons to the cuticle and displaces the ALM axons toward the dorsal midline and away from their wild-type position near the lateral midline.) Figure 3 (and its supplements) show AVM and ALM curvature in *unc-70(e524)* and *unc-70(e524);him-4* double mutants, illustrating that buckling defects similar to those seen previously only in AVM (Krieg, Dunn, & Goodman, Nat Cell Biol 2014) can occur in ALM neurons displaced towards the dorsal midline by loss of *him-4* function. These results suggest that AVM and ALM are both sensitive to decreased axial tension and mechanical compression. Finally, we revised the schematic diagrams to more clearly illustrate axon position and note that the still images shown in Figure 2 and Figure 3 are drawn from the supplementary videos.

*5) Figure 3: These data suggest that defects in spectrin reduce microtubule length in the ALM, possibly due to fracture. However, it also shows that this effect can be rescued by expressing spectrin in the epidermis, i.e. non-cell autonomously. This suggests that axial stress is rescued in this condition. It would be very good to test that by STED imaging of unc-70(e524) in the presence of epidermal unc-70.*

This comment concerns our observation that microtubule length is decreased in *unc-70(e524)* mutants compared to wild-type and that expressing wild-type UNC-70 β–spectrin in the epidermis rescues (partially) this effect (Figure 9). We agree that these observations point toward a non-cell autonomous effect and added this text to communicate this interpretation:

“Together, these findings suggest that the longer ALM MTs are more susceptible to bending-induced fracture than the shorter AVM MTs. In addition, an analysis of MT length and number in β-spectrin mutants suggests that the actin-spectrin network provides a lateral support that protects the MT bundle against deformation via both cell-autonomous and non-autonomous mechanical stabilization.”

The reviewers also infer that expressing wild-type UNC-70 β-spectrin in the epidermis restores axial pre-stress (tension) to *unc-70(e524)* mutants and suggest testing this idea by STED imaging. To achieve this goal, we would need to visualize spectrin periodicity in vivo. Technical barriers prevent us from conducting this experiment. The tag we used to visualize β-spectrin would rescue *unc-70(e524)* in all tissues since it is a full-length fusion [N-TSMod-UNC-70(+), previously reported in Krieg, Dunn & Goodman, 2014] expressed under the endogenous *unc-70* promoter. Thus, we cannot perform STED imaging of β -spectrin in the mutant background either in vivo or in vitro. We used α-spectrin SPC-1::GFP to visualize spectrin periodicity in wild-type and *unc-70(e524)* dissociated neurons in culture (Figure 5). Our efforts to analyze α-spectrin periodicity in vivo were unsuccessful – which is why we performed STED imaging experiments on cultured neurons. We suspect that the expression of SPC-1::GFP in the epidermis interferes with our ability to detect periodicity in the spectrin distribution in the TRNs, despite the increased spatial resolution afforded by STED imaging.

*In addition, do the axons still buckle when unc-70 is expressed in the epidermis? Pictures/analysis as in Figure 2 should be shown for this situation.*

Data addressing this comment were shown previously (Krieg, Dunn, and Goodman, Nat Cell Biol, 2014)—*unc-70* mutants expressing the epi::UNC-70(+) transgene have reduced axon buckling compared to *unc-70* mutants lacking this transgene. As a result, we chose not to reprise these findings in the present manuscript.

*Further, the authors show that epidermis expression of UNC-70 partially rescues the microtubule number and length in ALM neuron in the null mutation unc-70(s1502) but not in the tetramer domain mutant unc-70(e524). Could the authors comment on that? An equally plausible explanation for that rescue could be that these animals are paralyzed, which as detailed in Hammarlund, Jorgensen and Bastiani, 2007) is sufficient to rescue some of the unc-70 axon phenotypes.*

This comment is related to the one above but includes a query about the differences between *s1502* null allele and the *e524* missense allele of *unc-70*. To reiterate, we observe that expression of the epi::UNC-70(+) transgene increases MT length and number in both *unc-70* alleles, but that the effect size is bigger in the *s1502* allele than in the *e524* mutant (Figure 9). We agree that the paralysis exhibited by *s1502* mutants, but not by *e524* mutants could be a factor such that the inability of *s1502* animals to move could better preserve MTs in this genetic background. We added this alternative interpretation to the text:

“Related to the expected mechanical stresses generated in these animals, *e524* animals have a loopy and uncoordinated locomotion phenotype and *s1502* animals are paralyzed (Hammarlund et al., 2000). […] In light of these differences, we interpret the increased variance in MT as an indication that some long MTs persist in *s1502* mutants because mechanical stresses are decreased in this genetic background and other MTs are shortened due to the loss of actin-spectrin networks.”

*Furthermore, why are MTs in the AVM not shorter upon unc-70 mutation, even though these cells experience more mechanical deformations (Figure 2)? Does a length-dependent persistence length along published values really explain this?*

We also found this observation puzzling at first. Upon reflection, the simplest interpretation is that both the present findings and the classical studies of Chalfie and Thompson (1979) establish that MTs in wild-type AVM neurons are much shorter than those found in ALM and that the short MTs are less sensitive to the compression-induced buckling present in *unc-70* mutants. We have revised the text to reflect this interpretation (subsection “The spectrin network stabilizes TRN microtubules against fracture”, second paragraph), included a quantification of the buckling wavelength (Figure 9—figure supplement 2), and removed the speculation about length-dependent persistence length.

*Strikingly, the average MT length in the AVM is also more than twice shorter than previously published. What is the explanation for this?*

Table 2 reports mean values for MT length in the AVM neurons from the prior work (Chalfie & Thomson, 1979) and our analysis, which are 9.4 and 4.1 µm, respectively. Given that both the prior study and our study depend on a relatively small number of datasets (three ssTEM datasets in each case) and that the prior study used chemical fixation methods and ours used high-pressure freezing and freeze substitution, there is insufficient information to determine whether the mean values are really different (statistically speaking) or if the disparity in mean length is due to differences in sample preparation or another unknown factor. Because of these uncertainties, we prefer to provide readers with both data sets without further comment.

*Panel D of Figure 3 misses significance testing.*

The data from Figure 3 are now shown in Figure 9, together with a statistical analysis, using a two-way ANOVA and posthoc tests. The results are indicated in Figure 9 and support an effect of cell-type and genotype on MT length and an effect of cell-type, but not genotype on MT number. We note, however, that this significance testing has limited value since there are less than five ssTEM datasets/cell or genotype.

*6) The periodicity of spectrin distribution is not very obvious* in vivo*. For instance, it is difficult to see α-spectrin periodicity in the ALM axon Figure 4—figure supplement 2, and the peak in the PSD graph is not obvious. Could the authors improve the data?*

We agree that spectrin periodicity is difficult to detect in vivo. This is because the ALM is engulfed by the epidermal cell and that both cells express SPC-1::GFP, which we used to visualize spectrin periodicity. We suspect that the two cells are too close to adequately image independent of one another, even with super-resolution methods. This complication was evident in the PSD (Original Figure 4—figure supplement 2) that is much noisier than equivalent data obtained for neurons in vitro (Original Figure 4). We attempted to circumvent the interference between the epidermal and neuronal structures by imaging β-spectrin (which is expressed at lower levels in the epidermal cells). In this revision, we decided to focus on the higher quality STED imaging obtained from dissociated cells in culture and removed the data obtained from fixed animals from the manuscript.

*Could the authors explain why there are two peaks in Figure 4—figure supplement 2?*

The referees raise an interesting point. The PSD in the original Figure 4—figure supplement 2 was calculated by transforming the entire length of one commissure. The two peaks evident in the PSD suggest that spectrin networks may reside in different mechanical states in vivo such that the network adopts distinct configurations. This observation deserves further investigation in a separate study that is beyond the scope of the present work. Thus, to keep the present study focused on sensory neurons, we have revised Figure 4—figure supplement 2 to including only images of β-spectrin in touch receptor neurons.

*Besides, how do the authors explain that the localization of α spectrin is not perturbed in the unc-70(e524) mutant?*

This comment relates to the α–spectrin periodicity analyzed in vitro (original Figure 4; revised Figure 5) and asks about the appearance of α–spectrin in *unc-70(e524)* missense and *unc-70(s1502)* null mutants. In particular, we show that *e524* mutants retain a periodic distribution of α–spectrin but that the *s1502* mutants do not. Because the *s1502* allele is null and is expected to lack b-spectrin, these observations imply that no periodic actin-spectrin network is formed in this mutant even α–spectrin protein is retained. Consistent with this idea, STED images in *s1502* mutants are very similar to those observed in cells treated with the actin depolymerization agent, Latrunculin A (Compare Figure 5).

Further, *unc-70(e524)* is a point mutation mapping to the linker between the 16^th^ spectrin repeat and the tetramerization domain. It is thus likely, that this defect does only partially interfere with spectrin assembly, but makes the network mechanically less rigid. Interestingly, we also observe a reduction in length of the periodic structure, indicative of a subtle, but consistent effect suggesting a change in mechanics.

*Does expression of epidermal UNC-70 in the epidermis restore spectrin periodicity in axons?*

This comment overlaps with point 3 above and could only be addressed by analyzing spectrin periodicity in the presence and absence of the *oxIs95* transgene to drive expression of wild-type UNC-70 in the epidermis in vivo. As we cannot resolve the periodicity of α-spectrin in intact animals reliably due to background fluorescence from neighboring tissues, we removed all STED images taken from fixed animal from the revised manuscript.

*7) Figure 6: The twisted microtubules in panel d are very striking. Since this is a very important aspect of the paper, it would be good if this observation can be quantitatively compared to the other conditions. What is the pitch of the twist?*

We agree with the referees that a quantitative description would be a terrifically useful addition to the study. Accordingly, we have now quantified the flow field between selected cross-sections showing a clear rotation around the central bundle axis (Figure 7 and Figure 7Video 17). We also added a detailed description of the methodology to the Methods section of the article.

*Could the authors explain in more detail how the angle between MTs was determined (a scheme would help), and how the twisting of the axon and MTs would arise?*

We added a schematic to Figure 6—figure supplement 2 describing how the angle between the MTs was determined. We also simulated a hexagonal array and measured the angles between these points to illustrate that these three angles neatly describe a hexagonal lattice. We added a text describing how the twisting of the MT bundle could arise, namely, due to a chiral torque originating from the interactions between helically grooved molecules which generically prefer a finite angle between neighboring filaments in a bundle (subsection “Microtubule bundles are disrupted in *mec-7(wy116)* β-tubulin and *ptl-1(ok621)* tau mutants”, fourth paragraph).

*Does the observed twist depend on the animal moving, and would it be observed in a fully paralyzed mutant?*

If the reviewers are referring to the twisted MT phenotype, it is detected in fixed samples (Figure 6—figure supplement 3; Figure 7). We also see plectonemes and coils in animals that have been constrained by agar embedding methods, in which animals cannot move and are immobilized.

*8) Figure 6: FRAP is discussed before a large fraction of Figure 6 is discussed. Please consider restructuring this part a bit. I.e. put 287-302 before the FRAP data.*

We reorganized the paragraph as suggested by the reviewers as part of the global reorganization.

*9) Figure 8: what happens in ptl-1 mutants that have epidermal expression of unc-70?*

*Further, laser cutting experiment is not at all convincing, since the torque release in the Video 8 and Figure 8 is difficult to see. Could the author provide clearer examples? Besides, the color-coded image is quite difficult to interpret. Please show separate stills from the movie and mention how often this was observed.*

This comment appears to encompass two subpoints. Firstis the question of whether or not epidermal expression of UNC-70(+) would affect *ptl-1* mutants. We note that *ptl-1* mutants express wild-type *unc-70* in neurons and epidermis—were the reviewers thinking instead of how epidermal expression of UNC-70(+) affects the plectonemes observed in *ptl-1(ok621);unc-70(e524)* double mutants? This is an interesting question, especially since we have previously shown that expressing UNC-70(+) partially suppresses movement-induced buckling in *unc-70(e524)* mutants [Krieg, Dunn & Goodman, Nat Cell Biol, 2014]. [To address this question, we created *ptl-1(ok621);unc-70(e524);oxIs95* [EPI::UNC-70(+)] triple mutants and imaged the resulting animals under the same conditions used to observe *ptl-1;unc-70* double mutants. We found that the epidermal re-expression of *unc-70* did not abolish the development of the plectonemes, although we observed a subtle effect on plectoneme size. We included these data in Figure 1.

Second, the reviewers did not find the laser axotomy experiment a convincing demonstration of torque present in *ptl-1;unc-70* double mutant axons and asked us to provide clearly examples and to improve the presentation of this result. We have now repeated these experiments and added new example videos to this manuscript, clear showing three dimensional re-arrangements and rotations (Figure 1Video 6). We also added stills of this video to the Figure 1. All together we performed a total of ~20 laser axotomies showing rearrangements of these defects.

*10) Data and Discussion on torque generation:*

*The authors present AFM data to show how stiffness is perturbed in unc-70 mutants or after Latrunculin A treatment. The parameter space derived from their model predicts that the formation of plectonemes through torque depends on tension and stiffness. To substantiate why axons do not form plectonemes, despite the appearance of torque in ptl-1(ok621), it would be important to determine the stiffness of ptl-1 axons and/or mec-7(wy116) mutants (prediction would be that it should be normal).*

We have now repeated the AFM experiments on the *ptl-1* and *mec-7* mutants and included these data in Figure 4—figure supplement 2. In contrast to the prediction, we found that these mutations have a lower Young’s modulus. Because they have a high stabilizing tension, we do not see formation of plectonemes. The *unc-70(e524);ptl-1(ok621)* double mutation, has a lower E as well as a low tension (strictly, we did not measure tension in the double mutation but infer that the effect of *unc-70(e524)* single mutation on tension is preserved in the double.

*The Results section on ptl-1 ends with a bold statement saying "organized into a twisted bundle, implying ptl-1 mutant axons have both decreased bending stiffness and increased torque". What is the basis for this statement? It would help to have in the text the constitutive equation accounting for it (if based on an equation).*

We have now included a paragraph relating the number and cross-section of the MTs within the bundle to its mechanics. We write to inform the reader that the anatomy and geometry of the bundle relates to its mechanics according to the references cited therein:

“Given that bundle stiffness is expected to be proportional to the number of filaments in the bundle (Bathe et al., 2008; Guo et al., 2007), this finding suggests that bending stiffness is decreased in both of these mutants. […] A decrease in area and in the number of cross-links is expected to decrease the bending stiffness of the bundle (Bathe et al., 2008; Tolomeo and Holley, 1997).”

*The origin of torque generation should be discussed better. The authors suggest that MTs twist to maximize the length of contact lines. This suggests that it is related to the microtubules having 15 protofilaments and therefore having a supertwist. Does the pitch of twisting correspond with the pitch of the protofilament supertwist? Does this mean that 13 pf do not show this effect and that these finding cannot be translated to mammalian axons?*

We have now included a discussion about 13-protofilament microtubules in the main text. In essence, we believe that our results translate to all MT, as every MT possess a helical surface lattice, independent of its protofilament number. We also added a citation, showing the development of similar coils in cultured, mammalian axons (Roland et al., 2014). We added the following sentence to the Discussion:

“As all microtubules possess a helical surface lattice (Hunyadi et al., 2007), this mechanism is independent of the number of protofilaments per microtubule and applies not only of the unusual 15-protofilament MTs present in *C. elegans* TRNs, but also to the more common 13-protofilament MTs found in mammalian neurons. Consistent with this idea, MTs bend and deform into similar structures under compressive forces in cultured hippocampal neurons (Roland et al., 2014) and form twisted, hexagonal bundles in certain in vitro preparations (Needleman et al., 2004).”

*11) Methods: It would be important to better explain in the main text how the shape analysis was quantified and done. The equation provided in the Methods is not very clear: (i) why sum from i=0 to i=N-n? (ii) my understanding of path length is that it measures a distance, and should thus be expressed in meters; if so, the <∆χ2> would overall be expressed in meters, correct?*

We have now included a detailed description of the shape quantification in the main text:

“To enable quantitative comparisons, we devised a method to quantify the neurite’s deviation from the straightest possible path[…] In this computation, a perfectly straight neuron has an effective constant D=0 and all deviations from straightness increase D, regardless of the nature of the morphological defect (see Methods).”

The equation and method used to quantify the randomness is conceptually very similar to the formalism to quantify the diffusion constant by calculating its mean squared displacement average over increasing time windows. In our calculation, instead of averaging over increasing time windows, we average over increasing spatial windows. The advantage of our formalism is that it is very sensitive to local deviations from a straight line (e.g. due to buckling and shape defect), but less sensitive to non-local deviations, e.g. neuron curvature due to changes in body posture.

We have also included a detailed description of the underlying mean squared deviation (MSD) formalism in the Methods (subsection “Axon shape analysis”). The units for the spatial MSD are length^2^/length, which means it is the scale of a length.